# PRECONDITIONING FOR PHYSICS-INFORMED NEURAL NETWORKS

## ABSTRACT

Physics-informed neural networks (PINNs) have shown promise in solving complex partial differential equations (PDEs). However, certain training pathologies have emerged, compromising both convergence and prediction accuracy in practical applications. In this paper, we propose to use condition number as an innovative metric to diagnose and rectify the pathologies in PINNs. Inspired by classical numerical analysis, where the condition number measures sensitivity and stability, we highlight its pivotal role in the training dynamics of PINNs. We delineate a theory that elucidates the relationship between reduced condition numbers and improved error control, as well as better convergence. Subsequently, we present an algorithm that leverages preconditioning to enhance the condition number. Evaluations on 18 PDE problems showcase the superior performance of our method. Significantly, in 7 of these problems, our method reduces errors by an order of magnitude. Furthermore, in 2 distinct cases, our approach pioneers a solution, slashing relative errors from roughly 100% to below 6% and 21%, respectively.

## 1 INTRODUCTION

Numerical methods, such as finite difference and finite element methods, discretize partial differential equations (PDEs) into linear equations to attain approximate solutions. Such discretizations can be computationally burdensome, especially for PDE-constrained problems that demand frequent solver calls. Recently, physics-informed neural network (PINN) (Raissi et al., 2019) and its extensions (Pang et al., 2019; Yang et al., 2021; Liu et al., 2022) have emerged as powerful tools for tackling these challenges. By integrating PDE residuals into the loss function, PINNs not only ensure that the neural network adheres to the physical constraints but also maintain its versatility for a spectrum of PDE-related problems, including inverse problems (Chen et al., 2020; Jagtap et al., 2022) and physics-informed reinforcement learning (PIRL) (Liu & Wang, 2021; Martin & Schaub, 2022). While PINNs have successes over various domains (Zhu et al., 2021; Cai et al., 2021; Huang & Wang, 2022), their full potential and capabilities remain under-explored.

Several studies (Mishra & Molinaro, 2022; De Ryck & Mishra, 2022; De Ryck et al., 2022; Guo & Haghighat, 2022) have theoretically demonstrated the proficiency of PINNs in addressing a vast majority of well-posed PDE problems. Yet, Krishnapriyan et al. (2021) spotlights the training *pathology* inherent to PINNs and shows their failure in even moderately complex problems[1] encountered in real-world scenarios. As illustrated in Figure 1, such pathology can substantially hinder convergence and prediction accuracy. While some researchers attribute the pathology to the unbalanced competition between PDE and boundary condition (BC) loss terms (Wang et al., 2021; 2022b), others advocate for enforcement of the BC on the neural network, eliminating BC terms altogether (Berg & Nyström, 2018; Sheng & Yang, 2021; Lu et al., 2021b; Sheng & Yang, 2022; Liu et al., 2022). However, this challenge persists as previous approaches only partially address the pathology when dealing with complex PDEs, such as the Navier-Stokes equations (Liu et al., 2022). Thus, effective strategies to address this pathology still remain largely open.

In this work, we introduce the condition number as a novel metric, motivated by its pivotal role in understanding computational stability and precision, to accurately quantify pathologies in PINNs.

---

[1]The term "complex problems" is employed here to describe PDEs characterized by nonlinearity, irregular geometries, multi-scale phenomena, or chaotic behaviors. For an in-depth discussion, see Hao et al. (2022).

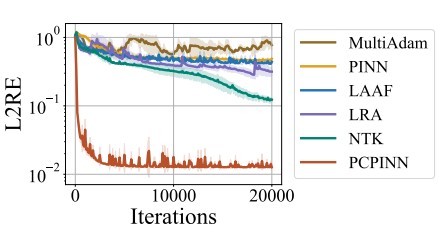 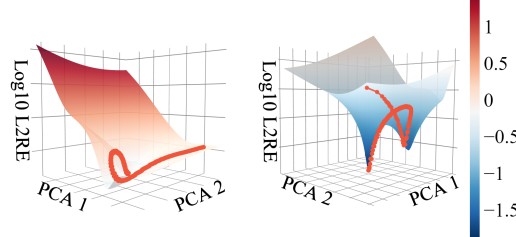

(a) Convergence dynamics: mean ± std      (b) Error landscape: PINN (left) vs. Ours (right)

Figure 1: An illustrative example of learning 1D wave equation. **(a)** PINN baselines (only a subset are shown) grapple with prolonged plateaus and severe oscillations during training. In contrast, our preconditioned PINN (PCPINN) can converge quickly and achieve much lower $L^2$ relative error (L2RE). **(b)** PINN wanders in the high-error zone (red), while ours dives deep and eventually converges. Scatters mark the model parameters in each iteration. Details are elaborated in Section 5.3.

Further, we present an algorithm specifically designed to optimize this metric, enhancing both accuracy and convergence. In traditional numerical analysis, the condition number acts as a beacon, highlighting the sensitivity of a function's output relative to its input. A higher condition number typically indicates potential issues, such as susceptibility to errors, often making algorithmic convergence challenging. This insight is particularly relevant in machine learning's complex optimization landscape. In this context, the condition number emerges as a vital tool to identify potential pitfalls. Given its profound significance in numerical analysis and its potential in machine learning, we highlight its promise for addressing intricacies inherent to PINNs.

Specifically, we first theoretically demonstrate that, under appropriate assumptions, a lower condition number correlates with improved error control and faster convergence. We then propose an algorithm that mitigates the condition number by incorporating a preconditioner into the loss function. To validate our theoretical framework, we evaluate our approach on selected PDE problems and further benchmark it against a comprehensive PINN dataset (Hao et al., 2023), which encompasses 20 distinct forward PDEs and 2 inverse scenarios. Our results consistently show state-of-the-art performance across most test cases. Notably, challenges previously deemed unsolvable for PINNs now become tractable, with dramatic error reductions seen in cases like a 3D Poisson equation with intricate geometry (reducing relative errors from nearly $100\%$ to below $5\%$).

## 2 PRELIMINARIES

We start by presenting the problem formulation and reviewing physics-informed neural networks (PINNs). We consider low-dimensional boundary value problems (BVPs) [2] that expect a solution $u$ satisfying that:

$$\mathcal{F}[u] = f \quad \text{in } \Omega, \tag{1}$$

with a boundary condition (BC) of $u|_{\partial\Omega} = g$, where $\Omega$ is an open, bounded subset of $\mathbb{R}^d$ with dimension $d \leq 4$. Here, $f: \Omega \to \mathbb{R}$ and $g: \partial\Omega \to \mathbb{R}$ are known functions; $\mathcal{F}: V \to W$ is a partial differential operator including at most $k$-order partial derivatives, where $k \in \mathbb{N}^+$ and $V, W$ are normed subspaces of $L^2(\Omega)$.

Assuming the well-posedness of our BVP, a fundamental property of formulations for physical problems, as indicated by Hilditch (2013), we can find a subspace $S \subset \mathcal{F}(V)$. For every $w \in S$, there exists a unique $v \in V$ such that $\mathcal{F}[v] = w$ and that $v|_{\partial\Omega} = g$, that is, the BC. This allows us to define $\mathcal{F}^{-1}: S \to V$ as $\mathcal{F}^{-1}[w] = v$. Again, owing to the well-posedness, $\mathcal{F}^{-1}$ is continuous within $S$. Conclusively, our solution can be framed as $u = \mathcal{F}^{-1}[f]$.

PINNs use a neural network $u_{\boldsymbol{\theta}}$ with parameters $\boldsymbol{\theta} \in \Theta$ to approximate the solution $u$, where $\Theta = \mathbb{R}^n$ represents the parameter space and $n \in \mathbb{N}^+$ is the number of parameters. The optimization

---

[2]Although not discussed, our method readily extends to problems involving vector-valued functions and more general boundary conditions. Relevant experimental details can be found in Appendix D.

problem of PINNs can be formalized as a constrained optimization problem:

$$\min_{\boldsymbol{\theta} \in \Theta} \|\mathcal{F}[u_{\boldsymbol{\theta}}] - f\|, \quad \text{subject to } u_{\boldsymbol{\theta}}|_{\partial\Omega} = g. \tag{2}$$

Two primary strategies to address the constraint are:

$$\mathcal{L}_{\text{soft}}(\boldsymbol{\theta}) = \|\mathcal{F}[u_{\boldsymbol{\theta}}] - f\|^2 + \alpha \|u_{\boldsymbol{\theta}} - g\|_{\partial\Omega}^2 \quad \text{or} \quad \mathcal{L}_{\text{hard}}(\boldsymbol{\theta}) = \|\mathcal{F}[\hat{u}_{\boldsymbol{\theta}}] - f\|^2, \tag{3}$$

where $\alpha \in \mathbb{R}^+$, $\|\cdot\|_{\partial\Omega}$ denotes the norm evaluated at $\partial\Omega$, and all the norms are estimated via Monte Carlo integration. The first approach adds a penalty term for BC enforcement. However, as highlighted by Wang et al. (2021), this can induce loss imbalances, leading to training instability. In contrast, the second approach, as advocated by (Berg & Nyström, 2018; Lu et al., 2021b; Liu et al., 2022), employs a specialized ansatz: $\hat{u}_{\boldsymbol{\theta}}(\boldsymbol{x}) = l^{\partial\Omega}(\boldsymbol{x})u_{\boldsymbol{\theta}}(\boldsymbol{x}) + g(\boldsymbol{x})$, with $l^{\partial\Omega}$ being a smoothed distance function to $\partial\Omega$. This method naturally adheres to the BC, mitigating potential imbalances. We favor this approach and, for clarity, will subsequently omit the hat notation, assuming $u_{\boldsymbol{\theta}}$ fulfills the BC.

**Training Pathology.** Despite the promise of hard-constraint methods, training divergence still occurs in moderately complex PDEs (Liu et al., 2022). As noted by Krishnapriyan et al. (2021), minor imperfectness during optimization can lead to an unexpectedly large error, substantially destabilizing training. Our subsequent analysis will delve into this pathology.

## 3 ANALYZING PINNs' TRAINING DYNAMICS VIA CONDITION NUMBER

### 3.1 INTRODUCING CONDITION NUMBER

In the realm of numerical algorithms, condition number has long been a touchstone for understanding the problem landscape (Süli & Mayers, 2003). For instance, in linear algebra, the condition number of a matrix provides insights into the error amplification and propagation from input to output, thus indicating potential stability issues. Furthermore, in deep learning, condition number can be used to characterize the sensitivity of the network prediction. A "sensitive" model could be vulnerable to some malicious adversarial noise (Beerens & Higham, 2023).

Drawing inspiration from the established knowledge, we propose to use condition numbers to quantify the pathology of PINNs, offering a fresh perspective on their behavior.

**Definition 3.1** (**Condition Number**). For the boundary value problem (BVP) in Eq. (1), denoted by $\mathcal{P}$, by assuming the neural network has sufficient approximation capability (see Assumption A.4), the *relative* condition number for solving $\mathcal{P}$ with a PINN is defined as:

$$\text{cond}(\mathcal{P}) = \lim_{\epsilon \to 0^+} \sup_{\substack{0 < \|\delta f\| \le \epsilon \\ \boldsymbol{\theta} \in \Theta}} \frac{\|\delta u\|/\|u\|}{\|\delta f\|/\|f\|}, \tag{4}$$

provided $\|u\| \ne 0$, $\|f\| \ne 0^3$, where $\delta u = u_{\boldsymbol{\theta}} - u$ and $\delta f = \mathcal{F}[u_{\boldsymbol{\theta}}] - f$.

*Remark.* The condition number signifies the asymptotic worst-case relative error in prediction for a relative error in optimization (noticing that $\mathcal{L}(\boldsymbol{\theta}) = \|\delta f\|^2$). The problem is said to be *ill-conditioned* if the condition number is large, indicating that a small optimization imperfectness can result in a large prediction error. This suggests that the problem is highly sensitive to errors, complicating the identification of the correct solution.

Aligning with the observation that most real-world physical phenomena exhibit smooth behavior with respect to their sources, we assume that $\mathcal{F}^{-1}$ is locally Lipschitz continuous and present the subsequent theorem.

**Theorem 3.2.** *If $\mathcal{F}^{-1}$ is $K$-Lipschitz continuous with $K \ge 0$ in some neighbourhood of $f$, we have:*

$$\text{cond}(\mathcal{P}) \le \frac{\|f\|}{\|u\|}K. \tag{5}$$

---

³If $\|u\| = 0$ or $\|f\| = 0$, we can similarly define the *absolute* condition number by removing the two terms.

*Proof.* We defer the proof to Appendix A.1. □

*Remark.* It is worth emphasizing that $K$ is fundamentally tied to the intrinsic nature of the problem, with its dependence on a specific algorithm being minimal. Consequently, algorithmic enhancements, whether in network architecture or training strategy, may not substantially mitigate the pathology unless the problem is reformulated.

For specific cases such as linear PDEs, we could have simpler conditions for the condition number's existence (refer to Appendix A.2).

To provide a more hands-on understanding, we delve into a foundational problem in both mathematics and physics, the Poisson equation. We consider the equation:

$$
\begin{aligned}
\Delta u(x) &= f(x), & x \in \Omega = (0, 2\pi/P), \\
u(x) &= 0, & x \in \partial\Omega = \{0, 2\pi/P\},
\end{aligned}
\tag{6}
$$

where $P$ is the system paratmer. Our objective is to draw insights about the condition number from this scenario. Proceeding, we derive an analytical expression for the condition number.

**Theorem 3.3.** *Consider the function spaces $V = H^2(\Omega)$ and $W = L^2(\Omega)$. Let $\mathcal{F}$ denote the Laplacian operator mapping from $V$ to $W$, i.e., $\mathcal{F} = \Delta : V \to W$. Define the inverse operator $\mathcal{F}^{-1} \colon \mathcal{F}(V) \to V$ such that for every $w \in \mathcal{F}(V)$, $\mathcal{F}^{-1}[w] = v$, where $v \in V$ is the unique function satisfying $\mathcal{F}[v] = w$ with boundary condition $v(0) = v(2\pi/P) = 0$. Then, the norm of $\mathcal{F}^{-1}$ is:*

$$
\|\mathcal{F}^{-1}\| = \frac{4}{P^2}.
\tag{7}
$$

*Proof.* For a detailed derivation, refer to Appendix A.3. □

In light of Proposition A.5, the condition number $\mathrm{cond}(\mathcal{P})$ for our problem is elegantly captured by $\frac{4\|f\|}{P^2\|u\|}$. Although this example is foundational, it illuminates the delicate interplay between system properties and condition numbers. As we transition to Section 5.2, we will delve deeper, exploring three practical problems and investigating how to numerically exstimate the condition number.

## 3.2 How Condition Number Affects Error & Convergence

Next, we will discuss the correlation between the condition number and the error control as well as the convergence rate of PINNs.

**Corollary 3.4 (Error Control).** *Assuming that $\mathrm{cond}(\mathcal{P}) < \infty$, there exists a function $\alpha \colon (0, \xi) \to \mathbb{R}, \xi > 0$ with $\lim_{x \to 0^+} \alpha(x) = 0$, such that for any $\epsilon \in (0, \xi)$,*

$$
\frac{\|u_{\boldsymbol{\theta}} - u\|}{\|u\|} \leq (\mathrm{cond}(\mathcal{P}) + \alpha(\epsilon)) \frac{\sqrt{\mathcal{L}(\boldsymbol{\theta})}}{\|f\|}, \quad \forall \boldsymbol{\theta} \in \Theta \wedge \sqrt{\mathcal{L}(\boldsymbol{\theta})} \leq \epsilon.
\tag{8}
$$

*Proof.* This theorem can be derived directly from Definition 3.1 (see Appendix A.4 for details). □

*Remark.* For well-posed BVPs, it is known that there is no error when the loss $\mathcal{L}(\boldsymbol{\theta})$ is precisely zero. However, the magnitude of the error remains elusive when $\mathcal{L}(\boldsymbol{\theta})$ is a small (but non-zero) value (due to optimization error). This theorem provides clarity by establishing an asymptotic relationship between the error and the loss function, where the condition number serves as a scaling factor. Consequently, improving the condition number emerges as a pivotal step to ensuring greater accuracy, as empirically validated in our ablation study (see Section 5.3).

Then, we will show how the condition number affects the convergence of PINNs. Firstly, we analyze the local convergence under the gradient descent:

$$
\boldsymbol{\theta}^{(k)} = \boldsymbol{\theta}^{(k-1)} - \eta \nabla_{\boldsymbol{\theta}} \mathcal{L}(\boldsymbol{\theta}^{(k-1)}), \quad k = 1, 2, 3, \ldots,
\tag{9}
$$

where $\eta > 0$ is a fixed learning rate. Assuming $\boldsymbol{\theta}^{(0)}$ gets close enough to a sufficiently accurate local minimum $\boldsymbol{\theta}^*$ (i.e., $\mathcal{L}(\boldsymbol{\theta}^*) \approx 0$), we can approximate $\mathcal{L}(\boldsymbol{\theta})$ using its truncated Taylor expansion, whose error is discussed in Appendix A.6. This tailored approach then allows us to treat the problem akin to a convex optimization scenario, paving the way for the subsequent theorem.

**Theorem 3.5** (**Local Convergence**). *Assuming that* $\mathrm{cond}(\mathcal{P}) < \infty$*, starting from* $\boldsymbol{\theta}^{(0)}$ *which locates in a small neighbourhood of* $\boldsymbol{\theta}^*$*, denoted by* $B(\boldsymbol{\theta}^*, r), r > 0$*, we run a gradient descent algorithm for* $k \geq 1$ *steps with a sufficiently small learning rate* $\eta$ *(see Assumption A.12), it follows that:*

$$\frac{\|u_{\boldsymbol{\theta}^{(k)}} - u\|}{\|u\|} \lessgtr \left(\mathrm{cond}(\mathcal{P}) + \alpha\left(\frac{1}{\sqrt{k}}\right)\right) \frac{\|\boldsymbol{\theta}^{(0)} - \boldsymbol{\theta}^*\|}{\sqrt{2\eta k}\|f\|}, \tag{10}$$

*where* $\|\cdot\|$ *is the* $L^2$ *vector norm and* $\alpha\colon (0, \xi) \to \mathbb{R}, \xi > 0$ *with* $\lim_{x\to 0^+} \alpha(x) = 0$.

*Proof.* The proof is deferred to Appendix A.5. □

Secondly, we discuss the global convergence of PINNs through the lens of the neural tangent kernel (NTK) theory (Jacot et al., 2018; Wang et al., 2022c). Considering an infinitesimally small learning rate, Eq. (9) becomes a continuous-time gradient flow:

$$\frac{\mathrm{d}\boldsymbol{\theta}}{\mathrm{d}t} = -\nabla\hat{\mathcal{L}}(\boldsymbol{\theta}), \quad t \in (0, +\infty) \wedge \boldsymbol{\theta}(0) = \boldsymbol{\theta}^{(0)}, \tag{11}$$

where $\hat{\mathcal{L}}$ is the loss function discretized on a set of collocation points $\{\boldsymbol{x}^{(i)}\}_{i=1}^N$ (refer to Appendix A.7). We derive the following theorem.

**Theorem 3.6** (**Global Convergence**). *Let* $U$ *be a set such that* $\{u_{\boldsymbol{\theta}(t)} \mid t \in [0, +\infty)\} \subset U$*. Suppose that* $\mathcal{F}^{-1}$ *is Fréchet differentiable in* $\mathcal{F}(U)$*. Under the assumption that* $\mathrm{cond}(\mathcal{P}) < \infty$ *and other assumptions in the NTK (Jacot et al., 2018; Wang et al., 2022c), the average convergence rate* $c(t)$ *at time* $t$ *(see Appendix A.7 for a detailed definition) satisfies that:*

$$c(t) \gtrless \underbrace{\frac{\|f\|^2/(\|u\|^2|\Omega|)}{(\mathrm{cond}(\mathcal{P}))^2 + \alpha(\mathcal{L}(\boldsymbol{\theta}(t)))}}_{\text{condition number and physics}} \underbrace{\left\|\frac{\partial u_{\boldsymbol{\theta}(t)}}{\partial \boldsymbol{\theta}}\right\|^2}_{\text{neural network}}, \tag{12}$$

*where* $\alpha\colon (0, \xi) \to \mathbb{R}, \xi > \max_{t\in[0,+\infty)} \mathcal{L}(\boldsymbol{\theta}(t))$ *with* $\lim_{x\to 0^+} \alpha(x) = 0$.

*Proof.* The complete proof is given by Appendix A.7. □

*Remark.* According to the above two theorems, a small condition number could greatly accelerate the convergence. We empirically validate this finding in Section 5.2.

## 4 TRAINING PINNs WITH A PRECONDITIONER

In this section, we present a preconditioning method tailored to improve the condition number inherent to the PDE problem addressed by PINNs. This method paves the way for superior convergence and accuracy.

**Discretization of PDEs.** We begin with well-posed linear BVPs defined on a rectangular domain $\Omega$, where the differential operator $\mathcal{F}$ is linear. We employ the finite difference method (FDM) to discretize the BVP on a $N$-point uniform mesh $\{\boldsymbol{x}^{(i)}\}_{i=1}^N$: $\boldsymbol{A}\boldsymbol{u} = \boldsymbol{b}$. Here, $\boldsymbol{A} \in \mathbb{R}^{N\times N}$ is an invertible sparse matrix, $\boldsymbol{u} = (u(\boldsymbol{x}^{(i)}))_{i=1}^N$[4], and $\boldsymbol{b} = (f(\boldsymbol{x}^{(i)}))_{i=1}^N$.

**Preconditioning Algorithm.** For slightly complex problems, the condition number may reach the level of $10^3$ (see Section 5.2). To improve it, a preconditioning algorithm is employed to compute a matrix $\boldsymbol{P}$ to construct an equivalent linear system: $\boldsymbol{P}^{-1}\boldsymbol{A}\boldsymbol{u} = \boldsymbol{P}^{-1}\boldsymbol{b}$. Prevalent preconditioning algorithms such as incomplete LU (ILU) factorization (i.e., $\boldsymbol{P} = \widehat{\boldsymbol{L}}\widehat{\boldsymbol{U}} \approx \boldsymbol{A}$, where $\widehat{\boldsymbol{L}}, \widehat{\boldsymbol{U}}$ are sparse invertible lower and upper triangular matrices, respectively) can reduce the condition number by several orders of magnitude while keeping the time cost much cheaper than solving $\boldsymbol{A}\boldsymbol{u} = \boldsymbol{b}$ (Shabat et al., 2018). This can be formulated as:

$$\mathrm{cond}(\mathcal{P}) \approx \frac{\|\boldsymbol{b}\|}{\|\boldsymbol{u}\|}\|\boldsymbol{A}^{-1}\| \longrightarrow \frac{\|\boldsymbol{P}^{-1}\boldsymbol{b}\|}{\|\boldsymbol{u}\|}\|\boldsymbol{A}^{-1}\boldsymbol{P}\| \approx \frac{\|\boldsymbol{A}^{-1}\boldsymbol{b}\|}{\|\boldsymbol{u}\|}\|\boldsymbol{A}^{-1}\boldsymbol{A}\| = 1, \tag{13}$$

where $\|\cdot\|$ is the $L^2$ vector/matrix norm. A detailed derivation is provided in Appendix B.1. Finally, we can train PINNs with precomputed preconditioners as displayed in Algorithm 1.

---

[4]To be precise, $\boldsymbol{u}$ only approximately equal the point values of $u$ due to the error of numerical shemes.

---

**Algorithm 1** Training PINNs with a preconditioner

---

1: **Input:** number of iterations $K$, mesh size $N$, learning rate $\eta$, and initial parameters $\boldsymbol{\theta}^{(0)}$
2: **Output:** trained parameters $\boldsymbol{\theta}^{(K)}$
3: Generate a mesh $\{\boldsymbol{x}^{(i)}\}_{i=1}^N$ for the problem domain $\Omega$
4: Assemble the linear system $\boldsymbol{A}, \boldsymbol{b}$, where $\boldsymbol{A}$ is a sparse matrix
5: Compute the preconditioner $\boldsymbol{P} = \widehat{\boldsymbol{L}}\widehat{\boldsymbol{U}}$ via ILU, where $\widehat{\boldsymbol{L}}, \widehat{\boldsymbol{U}}$ are both sparse matrices
6: **for** $k = 1, \dots, K$ **do**
7:     Evaluate the neural network $u_{\boldsymbol{\theta}^{(k-1)}}$ on mesh points to obtain: $\boldsymbol{u}_{\boldsymbol{\theta}^{(k-1)}} = (u_{\boldsymbol{\theta}^{(k-1)}}(\boldsymbol{x}^{(i)}))_{i=1}^N$
8:     Compute the loss function $\mathcal{L}^\dagger(\boldsymbol{\theta}^{(k-1)})$ by:

$$\mathcal{L}^\dagger(\boldsymbol{\theta}) = \left\| \boldsymbol{P}^{-1}(\boldsymbol{A}\boldsymbol{u}_{\boldsymbol{\theta}} - \boldsymbol{b}) \right\|^2 = \left\| \widehat{\boldsymbol{U}}^{-1}\widehat{\boldsymbol{L}}^{-1}(\boldsymbol{A}\boldsymbol{u}_{\boldsymbol{\theta}} - \boldsymbol{b}) \right\|^2, \tag{14}$$

    which incorporates the following steps:
      (a)    Compute the residual $\boldsymbol{r} \leftarrow \boldsymbol{A}\boldsymbol{u}_{\boldsymbol{\theta}^{(k-1)}} - \boldsymbol{b}$
      (b)    Solve $\widehat{\boldsymbol{L}}\boldsymbol{y} = \boldsymbol{r}$ and let $\boldsymbol{r} \leftarrow \boldsymbol{y}$, which should be very fast since $\widehat{\boldsymbol{L}}$ is sparse
      (c)    Solve $\widehat{\boldsymbol{U}}\boldsymbol{y} = \boldsymbol{r}$ and let $\boldsymbol{r} \leftarrow \boldsymbol{y}$
      (d)    Compute $\mathcal{L}^\dagger(\boldsymbol{\theta}^{(k-1)}) = \|\boldsymbol{r}\|^2$
9:     Update the parameters via gradient descent: $\boldsymbol{\theta}^{(k)} \leftarrow \boldsymbol{\theta}^{(k-1)} - \eta \nabla_{\boldsymbol{\theta}}\mathcal{L}^\dagger(\boldsymbol{\theta}^{(k-1)})$
10: **end for**
    **Note:** In our implementation, there is no requirement to design a hard-constraint ansatz for $u_{\boldsymbol{\theta}}$ to adhere to the boundary conditions (BC). This is because our linear equation $\boldsymbol{A}\boldsymbol{u} = \boldsymbol{b}$ inherently encompasses the BC. Further details can be found in Appendix B.2.

---

**Time-Dependent & Nonlinear Problems.** While our primary focus in this section revolves around linear and time-independent PDEs, our approach is readily extended to handle both time-dependent and nonlinear problems with judicious adaptations. For time-dependent cases, there are strategies like treating time as an additional spatial dimension or a time-stepping iterative approach. As for nonlinear problems, techniques involve moving nonlinear terms to the bias $\boldsymbol{b}$ or utilizing iterative methods such as the Newton-Raphson method. We have elaborated on these adaptation strategies in Appendix B.3 for further reading.

**Non-Uniform Mesh & Broader Numerical Schemes.** While we employed the FDM with a uniform mesh for the sake of a streamlined formulation, it is essential to emphasize that this choice does not restrict our method's adaptability. In our implementation, we leverage more sophisticated numerical schemes, such as the finite element method (FEM) paired with a non-uniform mesh. To align the theory with this implementation, certain definitions, including norms, may necessitate subtle adjustments. For instance, a non-uniform mesh might demand a norm definition like $\| \cdot \| = (\int_\Omega |w(\boldsymbol{x}) \cdot (\cdot)|^2 \, \mathrm{d}\boldsymbol{x})^{\frac{1}{2}}$, where $w\colon \Omega \to \mathbb{R}$ represents a reweighting function.

## 5   Numerical Experiments

### 5.1   Overview

In this section, we design numerical experiments to address the following key questions:

- **Q1:** How can we calculate the condition number, and can it highlight pathology affecting PINNs' convergence and accuracy?

  In Section 5.2, we propose two estimation methods, validated on a problem with a known analytic condition number. We then apply these methods to approximate the condition number for three practical problems and study its relationship to PINNs' performance. Our results underscore a strong correlation, indicating the correctness of our theory.

- **Q2:** Can the proposed preconditioning algorithm improve the pathology, thereby boosting the performance in solving PDE problems?

  In Section 5.3, we evaluate our preconditioned PINN (PCPINN) on a comprehensive PINN benchmark (Hao et al., 2023) encompassing 20 PDEs from diverse fields. Employing the

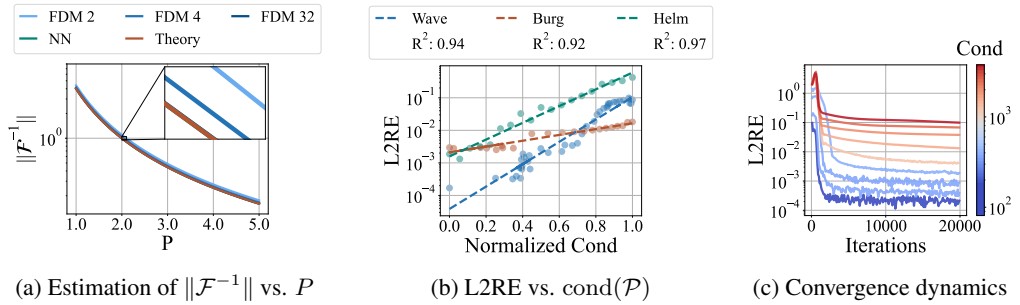

Figure 2: **(a):** Estimations of $\|\mathcal{F}^{-1}\|$ across different $P$ values, with the number after "FDM" indicating the mesh size. **(b):** Strong linear correlation between normalized condition numbers and associated errors. **(c):** Convergence in the wave equation across different condition numbers.

$L^2$ relative error (L2RE) as a primary metric (and MSE, L1RE as auxiliary ones), our approach sets a new benchmark: it reduces the error for 7 problems by a magnitude and renders 2 previously unsolvable (L2RE $\approx 100\%$) problems solvable.

- **Q3:** Does our method require extensive computation time?

  Figure 3a demonstrates that our approach is on par with PINN in terms of computational efficiency and even outpaces it in several instances. Furthermore, while Figure 3b indicates that neural network-based methods might not yet rival traditional solvers, they exhibit pronounced benefits in scalability. This suggests potential significant speed advantages for neural networks when addressing even larger problems.

Besides, the inverse problems are also studied (see Appendix D.3), and the supplementary experimental materials are deferred in Appendix C, D, and Appendix E.

## 5.2 RELATIONSHIP BETWEEN CONDITION NUMBER AND ERROR & CONVERGENCE

In this section, we empirically validate the theoretical findings in Section 3, especially the role of condition number in shaping the error and convergence dynamics of PINNs. Details of PDEs and implementation can be found in Appendix C. All experimental results are the average of 5 trials.

We begin by introducing two pragmatic techniques to estimate the condition number when provided the ground-truth solution:

1. Training a neural network to find the suprema in Eq. (4) with a small fixed $\epsilon$;
2. Leveraging the finite difference method (FDM) to discretize the PDEs and subsequently approximating the condition number using the matrix norm as discussed in Eq. (13).

To substantiate the reliability of these estimation techniques, we reconsider the 1D Poisson equation presented in Section 3.1. With $\|u\|$ and $\|f\|$ computed straightforwardly, our focus pivots to approximating $\|\mathcal{F}^{-1}\|$. Figure 2a captures our estimations across varied $P$ values, showcasing the close alignment with our theorem.

Transitioning to more intricate scenarios, we probe 3 practical problems: wave, Helmholtz, and Burgers' equation. System parameters within each problem are different: frequency $C$ in Wave, source term parameter $A$ in Helmholtz, and viscosity $\nu$ in Burgers. We vary the system parameter and monitor the subsequent influence on the condition number and error.

Figure 2b unveils a *strong*, but *simple* linear correlation emerges between normalized condition numbers and their corresponding errors, suggesting that the condition number could be a predictor for PINN performance. This relationship, however, varies depending on the normalization techniques used across different equations. For instance, in the wave equation, $\log(\text{L2RE})$ exhibits a linear relationship with $\log(\text{cond}(\mathcal{P}))$, while in Helmholtz, $\log(\text{L2RE})$ aligns with $\sqrt{\text{cond}(\mathcal{P})}$. A detailed interpretation of these patterns, through the lens of physics, is discussed in Appendix C.4. Lastly, Figure 2c underscores the condition number's profound impact on convergence trajectories, particularly evident in the wave equation, affirming the validity of our theoretical frameworks.

Table 1: Summary of the benchmark challenges. A "✓(*)" denotes that all problems in the category have the property. Otherwise, it is specific to the listed problems.

| Problem | Time-Dependency | Nonlinearity | Complex Geometry | Multi-Scale | Discontinuity | High Frequency |
|---|---|---|---|---|---|---|
| Burgers[1~2] | ✓(*) | ✓(*) | ✗ | ✗ | ✗ | ✓(2) |
| Poisson[3~6] | ✗ | ✗ | ✓(3 ∼ 5) | ✓(6) | ✓(5, 6) | ✗ |
| Heat[7~10] | ✓(*) | ✓(10) | ✓(9) | ✓(7, 8, 10) | ✗ | ✓(8) |
| NS[11~13] | ✓(*) | ✓(*) | ✓(12) | ✓(13) | ✗ | ✗ |
| Wave[14~16] | ✓(*) | ✗ | ✗ | ✓(16) | ✗ | ✓(15) |
| Chaotic[17~18] | ✓(*) | ✓(*) | ✗ | ✓(*) | ✗ | ✓(*) |

Table 2: Comparison of the average L2RE over 5 trials between our method and top PINN baselines. Best results are highlighted in blue and second-places in lightblue. "NA" denotes non-convergence or unsuitability for a given case. "⋆" signifies our method outperforming others by an order of magnitude or being the sole method to notably bring error under $100\%$.

| L2RE↓ | | Ours | Vanilla | | Loss Reweighting | | Optim | Loss Fn | Architecture | | |
|---|---|---|---|---|---|---|---|---|---|---|---|
| | | | PINN | PINN-w | LRA | NTK | MAdam | gPINN | LAAF | GAAF | FBPINN |
| Burgers | 1d-C | **1.42e-2** | 1.45e-2 | 2.63e-2 | 2.61e-2 | 1.84e-2 | 4.85e-2 | 2.16e-1 | 1.43e-2 | 5.20e-2 | 2.32e-1 |
| | 2d-C | 5.23e-1 | 3.24e-1 | 2.70e-1 | **2.60e-1** | 2.75e-1 | 3.33e-1 | 3.27e-1 | 2.77e-1 | 2.95e-1 | NA |
| Poisson | 2d-C⋆ | **3.98e-3** | 6.94e-1 | 3.49e-2 | 1.17e-1 | 1.23e-2 | 2.63e-2 | 6.87e-1 | 7.68e-1 | 6.04e-1 | 4.49e-2 |
| | 2d-CG⋆ | **5.07e-3** | 6.36e-1 | 6.08e-2 | 4.34e-2 | 1.43e-2 | 2.76e-1 | 7.92e-1 | 4.80e-1 | 8.71e-1 | 2.90e-2 |
| | 3d-CG⋆ | **4.16e-2** | 5.60e-1 | 3.74e-1 | 1.02e-1 | 9.47e-1 | 3.63e-1 | 4.85e-1 | 5.79e-1 | 5.02e-1 | 7.39e-1 |
| | 2d-MS⋆ | **6.40e-2** | 6.30e-1 | 7.60e-1 | 7.94e-1 | 7.48e-1 | 5.90e-1 | 6.16e-1 | 5.93e-1 | 9.31e-1 | 1.04e+0 |
| Heat | 2d-VC⋆ | **3.11e-2** | 1.01e+0 | 2.35e-1 | 2.12e-1 | 2.14e-1 | 4.75e-1 | 2.12e+0 | 6.42e-1 | 8.49e-1 | 9.52e-1 |
| | 2d-MS | **2.84e-2** | 6.21e-2 | 2.42e-1 | 8.79e-2 | 4.40e-2 | 2.18e-1 | 1.13e-1 | 7.40e-2 | 9.85e-1 | 8.20e-2 |
| | 2d-CG | **1.50e-2** | 3.64e-2 | 1.45e-1 | 1.25e-1 | 1.16e-1 | 7.12e-2 | 9.38e-2 | 2.39e-2 | 4.61e-1 | 9.16e-2 |
| | 2d-LT⋆ | **2.11e-1** | 9.99e-1 | 9.99e-1 | 9.99e-1 | 1.00e+0 | 1.00e+0 | 1.00e+0 | 9.99e-1 | 9.99e-1 | 1.01e+0 |
| NS | 2d-C | **1.28e-2** | 4.70e-2 | 1.45e-1 | NA | 1.98e-1 | 7.27e-1 | 7.70e-2 | 3.60e-2 | 3.79e-2 | 8.45e-2 |
| | 2d-CG | **6.62e-2** | 1.19e-1 | 3.26e-1 | 3.32e-1 | 2.93e-1 | 4.31e-1 | 1.54e-1 | 8.24e-2 | 1.74e-1 | 8.27e+0 |
| | 2d-LT | **9.09e-1** | 9.96e-1 | 1.00e+0 | 1.00e+0 | 9.99e-1 | 1.00e+0 | 9.95e-1 | 9.98e-1 | 9.99e-1 | 1.00e+0 |
| Wave | 1d-C | **1.28e-2** | 5.88e-1 | 2.85e-1 | 3.61e-1 | 9.79e-2 | 1.21e-1 | 5.56e-1 | 4.54e-1 | 6.77e-1 | 5.91e-1 |
| | 2d-CG | **5.85e-1** | 1.84e+0 | 1.66e+0 | 1.48e+0 | 2.16e+0 | 1.09e+0 | 8.14e-1 | 8.19e-1 | 7.94e-1 | 1.06e+0 |
| | 2d-MS⋆ | **5.71e-2** | 1.34e+0 | 1.02e+0 | 1.02e+0 | 1.04e+0 | 1.01e+0 | 1.02e+0 | 1.06e+0 | 1.06e+0 | 1.03e+0 |
| Chaotics | GS | **1.44e-2** | 3.19e-1 | 1.58e-1 | 9.37e-2 | 2.16e-1 | 9.37e-2 | 2.48e-1 | 9.47e-2 | 9.46e-2 | 7.99e-2 |
| | KS | **9.52e-1** | 1.01e+0 | 9.86e-1 | 9.57e-1 | 9.64e-1 | 9.61e-1 | 9.94e-1 | 1.01e+0 | 1.00e+0 | 1.02e+0 |

Abbreviations: "Optim" for optimizer, "MAdam" for MultiAdam, and "Loss Fn" for "Loss Function".

## 5.3 BENCHMARK OF FORWARD PROBLEMS

We delve into the comprehensive PINN benchmark, PINNacle (Hao et al., 2023), encompassing 20 forward PDE problems and 10+ state-of-the-art PINN baselines. These problems, highlighted in Table 1, span complexities from multi-scale behaviors to intricate geometries and diverse domains from fluids to chaos, underscoring the benchmark's depth and width. Further details can be gleaned from (Hao et al., 2023).

**Results and Performance.** From the suite of 20, we appraised our method on 18 problems, sidelining high-dimensional PDEs due to our method's mesh-based inherency. The experimental results are derived from 5 trials, with baseline outcomes sourced directly from the PINNacle paper. In most cases, as detailed in Table 2, our method emerged superior, showcasing a remarkable error drop (by an order of magnitude) for **7** problems. In **2** of these, ours uniquely achieved feasible solutions, with competitors approximating errors at nearly $100\%$. Our success is attributed to the employed preconditioner, mitigating intrinsic pathologies and enhancing PINN performance. For the supplementary results and experimental details, including PDEs, baselines, and implementation specifics, please refer to Appendix E and Appendix D.

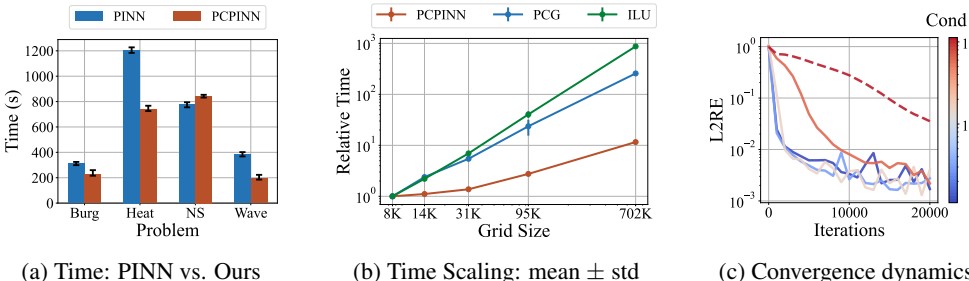

(a) Time: PINN vs. Ours     (b) Time Scaling: mean $\pm$ std     (c) Convergence dynamics

Figure 3: **(a):** Computation time of PCPINN (ours) and vanilla PINN in selected problems, with error bars showing the $[\min, \max]$ in 5 trials. **(b):** Scaling of computational time relative to an 8K grid size, contrasting our PCPINN with the preconditioned conjugate gradient method (PCG) and the preconditioning (ILU). **(c):** Convergence under varying preconditioner precision, with the dashed line for no preconditioner and the color bar for resultant condition numbers $\frac{\|\boldsymbol{P}^{-1}\boldsymbol{b}\|}{\|\boldsymbol{u}\|}\|\boldsymbol{A}^{-1}\boldsymbol{P}\|$.

**Convergence Analysis.** Using the 1D wave equation for illustration, our method's convergence dynamics surpass those of traditional baselines. As depicted in Figure 1a, we achieve *superexponential* convergence, while baselines show a slower, fluctuating trajectory. Notably, their fluctuations look smaller than real because of the logarithm-scale vertical axis. This stark difference is further accentuated in Figure 1b, where our method swiftly identifies the correct minimum, attributed to our preconditioner's ability to reshape the optimization landscape, facilitating rapid convergence with minimal oscillations.

**Computation Time Analysis.** We contrast the computation time of our method with that of the vanilla PINN across diverse problems including Wave1d-C, Burgers1d-C, Heat2d-VC, and NS2d-C. As evident in Figure 3a, our method is efficient, sometimes even outpacing the baseline. This efficiency is largely due to our rapid preconditioner calculation (basically less than 3s) and avoidance of time-intensive automatic derivation. Furthermore, we assessed the scalability of our method, the conjugate gradient method (used by the FEM solver), and the ILU for large-scale problems like Poisson3d-CG. While the neural network currently lags behind traditional methods in speed, its growth rate is remarkably slower by nearly two orders of magnitude. As Figure 3b suggests, we anticipate superior scaling in even larger problems, thanks to the neural network's capacity to operate on low-dimensional manifolds, effectively mitigating the curse of dimensionality.

**Ablation Study.** In our approach, the pivotal factor is the precision of the preconditioner, measured by the deviation between $\boldsymbol{P}$ and $\boldsymbol{A}$. It is controlled by the drop tolerance in ILU. We conducted ablation studies on this specific parameter across four Poisson equation problems. Figure 3c depicts the convergence trajectory of our approach under varying condition numbers after preconditioning in Poisson2d-C. The outcomes indicate a gradual performance decline of our method with decreasing precision of the preconditioner. Absent a preconditioner, our method reverts to a PINN with a discrete loss function, consequently failing to achieve convergence. This underscores the indispensable role of the preconditioner in enhancing the efficacy of PINNs, reinforcing our theoretical stance that the condition number profoundly impacts convergence. Comprehensive experimental details are available in Appendix D.4.

## 6   Conclusion

In this work, we have spotlighted the central role of the condition number in deciphering the challenges inherent to PINNs, establishing it as a novel metric for training pathology. By weaving together insights from traditional numerical analysis with modern machine learning techniques, we have demonstrated a direct correlation between a reduced condition number and improved PINN performance. Our proposed algorithm, benchmarked against a comprehensive dataset, offers significant advancements in both accuracy and convergence, surmounting challenges previously deemed intractable. This study not only marks a pivotal stride in PINN research but also sets a promising trajectory for future endeavors in physics-informed machine learning.

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
