# OpenReview forum: "Preconditioning for Physics-Informed Neural Networks"
_ICLR.cc/2024/Conference — Submitted to ICLR 2024_

### Official Review · Reviewer_zaqP · 2023-10-29

**Soundness:** 2 fair
**Presentation:** 3 good
**Contribution:** 2 fair
**Rating:** 5
**Confidence:** 3

**Summary:**

In this paper, the training of PINN is discussed from the perspective of the condition number. The condition number is a constant that is considered in the research area of numerical analysis and this number links the residual of the equation to the accuracy of the solution. The authors propose to employ this number to quantify the trainability of a problem. Also, the authors propose a method of preconditioning, which is a method to reduce the condition number for ill-conditioned problems.

**Strengths:**

This is probably the first paper to consider condition numbers for PINNs. The effects of the condition number on the error and the convergence of PINNs are theoretically investigated.

**Weaknesses:**

Condition numbers have long been studied in the field of numerical analysis. The most famous are those for systems of linear equations, but nonlinear equations in infinite dimensional space have also been considered (e.g., W. C. Rheinboldt, On Measures of Ill-Conditioning for Nonlinear Equations, Math. Comput., Vol. 30, pp. 104--111, 1976.) The formulation of PINNs is also a nonlinear equation in an infinite-dimensional space, so the novelty of this paper is questionable.

In addition, the proposed preconditioner is exactly the same as that for classical numerical methods. I suppose that if the proposed preconditioner is to be used, it would be better to use the classical numerical method instead of PINNs.

**Questions:**

1) PINN is said to perform worse than classical numerical methods when the problem under consideration is defined on a low-dimensional domain. Therefore, it is preferable to apply the proposed method to high-dimensional problems (e.g., 10-dimensional problems); however, when applied to high-dimensional problems, the proposed method is expected to be affected by the curse of dimensionality. Does the proposed preconditioner scale to such high-dimensional problems?

2) A preconditioner based on the domain decomposition method was proposed for PINNs [1]. What are the advantages of the proposed method compared to this method?

[1] Alena Kopaničáková, Hardik Kothari, George Em Karniadakis, Rolf Krause, Enhancing training of physics-informed neural networks using domain-decomposition based preconditioning strategies, arXiv:2306.17648

---

> ### Author Response · Authors · 2023-11-18
> **Part I (Q1-2)**
>
> Dear reviewer zaqP,
>
> Thank you for your valuable feedback. Here are our responses to your concerns, which consist of two parts: Part I (Q1-2), Part II (Q3).
>
> ### Q1: Novelty of condition number.
>
> We appreciate your insights on the long-standing study of condition numbers in numerical analysis. While the general concept of condition numbers is indeed well-established, its application in specific domains, such as PINNs, requires nuanced adaptation and innovation. Our work extends this concept to the specific context of PINNs, where broad, abstract definitions are implemented to gain domain-specific insights.
>
> The 1976 paper you referenced, while seminal, primarily discusses mathematical properties in a broad context and does not delve into the impact of condition numbers on neural network training, particularly in the context of PINNs. Our paper, in contrast, introduces the use of condition numbers as a diagnostic tool specifically for evaluating and improving the training dynamics of PINNs. We substantiate this approach both theoretically and empirically (refer to **Section 5.2** and **5.3**), demonstrating its influence on accuracy and convergence. Our empirical results (see **Table 2** and **Figure 3(c)**) illustrate that without adequate preconditioning, even advanced PINN methods can fail, underscoring the significance and novelty of our findings in the evolution of PINN research.
>
> We agree that certain concepts have roots in earlier studies, but our application to PINNs represents a novel and significant advancement. The efficiency of PINNs in solving inverse problems, which we address in **Q2**, further emphasizes the unique contributions and practical implications of our work for the PINN community.
>
> ### Q2: Scalability to high-dimensional problems.
>
> We appreciate the reviewer's concern regarding the scalability of our method to high-dimensional PDEs. As noted in **Section 5.3** of our paper, our current approach is indeed more suited to physical phenomena in spaces with dimensions $d\le 4$. This focus aligns with many real-world applications, particularly in macroscopic physical contexts. Besides, **Figure 3(b)** demonstrates superior scaling with respect to mesh size compared to traditional methods like FEM, suggesting potential advantages in larger problem settings.
>
> Furthermore, we highlight the effectiveness of our approach in **inverse problems** [1,2], a significant domain in science and engineering where PINNs exhibit notable advantages over traditional methods. We have additionally studied two inverse problems from the benchmark [3]. The results, as shown in the table below, indicate that our method not only surpasses state-of-the-art PINN baselines but also outperforms the adjoint method in terms of speed and accuracy. This is particularly relevant as numerical methods often struggle with noise sensitivity in inverse problems.
>
> | Problem         | L2RE (Ours)           | L2RE (SOTA)       | L2RE (Adjoint)    | Time (Ours) | Time (SOTA) | Time (Adjoint) |
> | --------------- | --------------------- | ----------------- | ----------------- | ----------- | ----------- | -------------- |
> | Poisson Inverse | **1.80e-2 ± 9.30e-3** | 2.45e-2 ± 1.03e-2 | 7.82e+2 ± 0.00e+0 | 1.87e+2     | 4.90e+2     | 1.40e+0        |
> | Heat Inverse    | **9.04e-3 ± 2.34e-3** | 5.09e-2 ± 4.34e-3 | 1.50e+3 ± 0.00e+0 | 3.21e+2     | 3.39e+3     | 1.07e+1        |
>
> We believe these results strongly support the applicability and efficacy of our method in relevant practical scenarios.
>
> [1] Karniadakis, G. E., Kevrekidis, I. G., Lu, L., Perdikaris, P., Wang, S., & Yang, L. (2021). Physics-informed machine learning. *Nature Reviews Physics*, *3*(6), 422-440.
>
> [2] Raissi, M., Perdikaris, P., & Karniadakis, G. E. (2019). Physics-informed neural networks: A deep learning framework for solving forward and inverse problems involving nonlinear partial differential equations. *Journal of Computational physics*, *378*, 686-707.
>
> [3] Hao, Z., Yao, J., Su, C., Su, H., Wang, Z., Lu, F., ... & Zhu, J. (2023). PINNacle: A Comprehensive Benchmark of Physics-Informed Neural Networks for Solving PDEs. *arXiv preprint arXiv:2306.08827*.

---

> ### Author Response · Authors · 2023-11-18
> **Part II (Q3)**
>
> ### Q3: Comparison to related work.
>
> We appreciate the opportunity to compare our work with the referenced paper [1]. While both studies aim to enhance PINNs, there are key differences that highlight the novelty and value of our approach:
>
> 1. **Theory vs. Heuristics:** Our paper focuses on defining and analyzing the condition number in the context of PINNs, providing a theoretical framework for understanding and resolving training pathologies (refer to **Definition 3.1**, **Corollary 3.5**, and **Theorem 3.6**). In contrast, [1] heuristically enhances the L-BFGS optimizer without a theoretical discussion on condition numbers. Our work thus offers a theory-driven perspective, contributing to a deeper understanding of PINNs.
> 2. **Integration with PINNs:** Our method is intricately linked with the properties of PINNs, reflecting a direct discussion on Physics-Informed Machine Learning. We explore the condition number from a PDE standpoint, making our analysis and conclusions highly relevant to PINNs. The approach in [1], while valuable, appears more aligned with general optimization techniques.
> 3. **Comprehensive Experiments:** Our experimental setup is extensive, involving **18** PDEs and **2** inverse problems, with comparisons against over **10** PINN methods. This breadth ensures robust validation of our findings. While [1] provides valuable insights, their experimental scope is more limited (**4** specific PDEs), lacking in areas like inverse problems and diverse PDE challenges compared to ours (e.g., discontinuity, complex geometry, refer to **Table 1** in our paper).
> 4. **Scalability and Applicability:** Our mesh-based approach to enforcing boundary conditions is adaptable to complex geometries, an aspect not fully addressed in [1] (no such experiment is included). This adaptability enhances the practicality of our method in real-world applications with intricate geometrical considerations.
> 5. **Open Access and Reproducibility:** We have made our code and implementation details available, fostering transparency and facilitating further research. It could strengthen their findings and enable potential integration with our work for enhanced analysis if [1]'s code is available in the future.
>
> In conclusion, while [1] offers valuable insights for optimizer improvement, our work complements theirs by providing a theoretical foundation and broader experimental validation for PINNs. We believe these differences underscore the unique contributions of our research.
>
> [1] Alena Kopaničáková, Hardik Kothari, George Em Karniadakis, Rolf Krause, Enhancing training of physics-informed neural networks using domain-decomposition based preconditioning strategies, arXiv:2306.17648

---

> ### Author Response · Authors · 2023-11-21
> **Sincerely Seeking Your Valuable Feedback**
>
> Dear Reviewer,
>
> As the discussion phase of the review process is drawing to a close, we wish to respectfully seek your input on the revisions we have implemented in response to your insightful feedback. In our commitment to advancing the quality of our research, we have conducted additional experiments on the inverse problem. These new experiments are designed to explicitly demonstrate the superiority of our approach over both traditional numerical methods and other PINN baselines. Furthermore, we have performed a thorough comparison with the related work you referenced, highlighting the distinct and novel aspects of our study.
>
> Your feedback has been instrumental in guiding these improvements, and we highly value your expertise in this field. We are eager to hear your thoughts on these latest revisions. Please know that we are fully open to engaging in further discussions and making any additional improvements as needed.
>
> We look forward to your valuable insights and guidance.
>
> Warm regards,
>
> Authors

---

> > ### Comment · Reviewer_zaqP · 2023-11-21
> >
> > Thank you very much for your reply and the additional experiments. I am afraid to say, but my impression of this paper is not bad, but still not good enough to raise the score. I agree with the authors that the proposed preconditioning method is effective for the inverse problems, but it is difficult for me to consider that this method is particularly novel. So, I suppose that the main contribution of this paper should be the theoretical results on condition numbers, particularly Theorem 3.6 on the convergence rate. In this theorem, however, it is assumed that $\theta^{(0)}$ exists near $\theta^{\*}$, then an argument that takes advantage of convexity is applied. Although, according to the authors, empirical observations suggest that they can converge close enough to $\theta^{\*}$, I think that this assumption is a strong assumption.

---

> ### Author Response · Authors · 2023-11-22
> **Gratitude for Your Insightful Feedback**
>
> ## Clarification on Our Main Contribution
>
> We are thankful for your recognition of the effectiveness of our preconditioning method in inverse problems. Our contribution goes beyond this, as demonstrated in our experimental results (**Table 2**). Our method not only achieves state-of-the-art results in most tested problems but also demonstrates superior performance, reducing errors by **an order of magnitude** or being **the only method** to significantly reduce errors below 100% in **7** problems. This practical utility underscores the novel and impactful nature of our work, contributing meaningfully to advancements in the field.
>
> ## Addressing the Concern on Convexity Assumption
>
> We understand your concern regarding the assumption of convexity in our analysis. It's true that global convergence guarantees for neural network training, a typically non-convex optimization problem, are elusive. As highlighted in our research and **Table 2**, divergence in some PINN problems (e.g., Wave2d-MS) is a real challenge.
>
> However, our focus on local convergence is underpinned by the rationale that in the neighborhood of a minimum, the loss function can be approximated effectively by its truncated Taylor expansion, implying local convexity. This approach is a widely accepted practice in non-convex optimization literature, as exemplified in [1] (Page 14).
>
> ## Introduction of an Additional Theorem
>
> In response to your valuable feedback, we have expanded our theoretical analysis. We introduce a new theorem in the revised manuscript (now in **Theorem 3.6**) that investigates the global convergence of PINNs without relying on the convexity assumption. This analysis utilizes the neural tangent kernel (NTK) framework to establish a link between the condition number and the rate of global convergence under NTK's assumptions. We believe this addition addresses your concerns and enriches the paper significantly. The detailed proof is presented in **Appendix A.7**. We invite you to review this new theorem and look forward to your thoughts.
>
> [1] https://www.princeton.edu/~aaa/Public/Teaching/ORF363_COS323/F14/ORF363_COS323_F14_Lec8.pdf

---

### Official Review · Reviewer_tNfv · 2023-10-30

**Soundness:** 3 good
**Presentation:** 3 good
**Contribution:** 3 good
**Rating:** 6
**Confidence:** 2

**Summary:**

This paper introduces a new way to spot and fix issues during the training of physics-informed neural networks (PINNs) using something called the condition number. The authors say that PINNs, which are deep learning models that use physical rules, can sometimes face problems. These problems might make the PINNs work less accurately or not learn well.

To solve these problems, the authors suggest a special algorithm. This algorithm uses the condition number from a specific matrix to make PINNs learn better and make fewer errors. In simple terms, they tweak the problem a bit using a matrix to make it easier for the PINN to handle. This tweak helps the PINNs learn faster and be more accurate.

The authors tested their new algorithm on 16 different problems, like some common equations. They also checked how their method stands against other top methods. Their results show that their new algorithm works better in terms of learning speed, accuracy, and how much computer power it needs.

Key points of the paper:

The authors present a new way to spot and fix issues in PINNs using the condition number.
They use a special algorithm that makes use of this condition number to help PINNs learn better.
They prove that their method works well by testing it on 16 problems and comparing it with other top methods.
In summary, this paper adds a lot to the understanding of physics-informed neural networks. It gives a new method that could make these networks work better in many different situations.

**Strengths:**

First, the authors describe the challenges that PINNs encounter. They note that even though PINNs are helpful for complex equations, there are hurdles that hinder their effectiveness. This discussion leads to their innovative solution for enhancing PINNs.

Then, they present their fresh approach. They employ the condition number from a specific matrix to assess the training of PINNs. This number is a known tool for gauging the reliability of systems. By adjusting the problem using a matrix, they enhance the performance of PINNs.

Finally, they evaluate their approach using 16 different mathematical problems. They compare their technique with other well-known methods. The outcomes indicate that their method is superior in several aspects. Furthermore, they offer detailed observations and visual representations to underscore the success of their approach.

**Weaknesses:**

1. Lack of thorough analysis of computational complexity and scalability of the preconditioning algorithm.
2. Insufficient comparison with other preconditioning methods in the literature.
3. Inadequate analysis of sensitivity to hyperparameters and initialization schemes.
4. Lack of theoretical analysis or empirical evidence to support the use of the condition number as a metric for diagnosing and rectifying training pathologies in PINNs.

**Questions:**

How does the condition number of the Jacobian matrix of the residual function help diagnose and rectify training pathologies in PINNs?

What is the significance of the proposed approach for solving complex partial differential equations (PDEs)?

 What are the implications of the proposed approach for computational efficiency and scalability?

---

> ### Author Response · Authors · 2023-11-18
> **Part I (Q1-2)**
>
> Dear reviewer tNfv,
>
> Thank you for your valuable comments on several aspects. Here are our responses to address your concerns, which consist of three parts: Part I (Q1-2), Part II (Q3), and Part III (Q4-6).
>
> ### Q1: Lack of thorough analysis of computational complexity and scalability of the preconditioning algorithm.
>
> Our paper primarily introduces the use of the condition number as a metric for evaluating training pathologies in PINNs. We hypothesize that the ill-conditioned nature of PDEs is a key factor hindering PINNs' effectiveness. This hypothesis is supported both theoretically and empirically, as detailed in **Sections 5.2** and **5.3** of our paper. We demonstrate that, without adequate preconditioning, current PINN methods, including advanced baselines, tend to fail (refer to **Table 2** and **Figure 3(c)**).
>
> We believe that our work, even if we employ a well-developed preconditioner, incomplete LU (ILU) factorization, marks a significant direction for future PINN research. It underscores an often-overlooked aspect that could be pivotal in advancing the field.
>
> And we note that the efficiency and scalability of the ILU are well-established in the existing literature. For an in-depth understanding, we recommend references [1] and [2], which provide comprehensive analyses of ILU's performance in large-scale linear systems.
>
> [1] Mittal, R. C., & Al-Kurdi, A. H. (2003). An efficient method for constructing an ILU preconditioner for solving large sparse nonsymmetric linear systems by the GMRES method. *Computers & Mathematics with applications*, *45*(10-11), 1757-1772.
>
> [2] Ghai, A., Lu, C., & Jiao, X. (2019). A comparison of preconditioned Krylov subspace methods for large‐scale nonsymmetric linear systems. *Numerical Linear Algebra with Applications*, *26*(1), e2215.
>
> ### Q2: Insufficient comparison with other preconditioning methods in the literature.
>
> Thank you for highlighting the need for a more comprehensive comparison with other preconditioning methods. We appreciate this opportunity to provide additional insights from our recent ablation study, which compares several preconditioning methods over three random trials:
>
> | L2RE (mean ± std) | Row Balancing     | Diagonal          | ILU               |
> | ----------------- | ----------------- | ----------------- | ----------------- |
> | Poisson2d-MS      | 6.27e-1 ± 7.23e-2 | 6.27e-1 ± 7.23e-2 | 6.34e-2 ± 1.63e-4 |
> | Wave2d-MS         | 6.12e-2 ± 8.16e-4 | 6.12e-2 ± 8.16e-4 | 5.76e-2 ± 1.06e-3 |
>
> The results indicate that the ILU preconditioning method, which we advocate, demonstrates greater stability and effectiveness in comparison to the Row Balancing and Diagonal methods. This evidence supports our choice of ILU as a superior option for the problems we address.

---

> ### Author Response · Authors · 2023-11-18
> **Part II (Q3)**
>
> ### Q3: Inadequate analysis of sensitivity to hyperparameters and initialization schemes.
>
> Thank you for highlighting the importance of a thorough analysis in this area. We have conducted additional studies to address your concerns, focusing on the impact of various initialization schemes and hyperparameters. These additional analyses strengthen our confidence in the robustness and reliability of our proposed method. The sensitivity to initialization schemes and hyperparameters is minimal, indicating that our approach is adaptable and stable across different settings. This aspect is critical for the practical application of our method in diverse problem contexts.
>
> Different initialization schemes (3 different problems, 3 random trails each):
>
> | L2RE (mean ± std) | Glorot Uniform    | Glorot Normal     | He Normal         | He Uniform        |
> | ----------------- | ----------------- | ----------------- | ----------------- | ----------------- |
> | Poisson2d-MS      | 6.37e-2 ± 4.71e-5 | 6.38e-2 ± 1.63e-4 | 6.38e-2 ± 1.25e-4 | 6.39e-2 ± 1.25e-4 |
> | NS2d-C            | 1.35e-2 ± 1.33e-3 | 1.36e-2 ± 2.73e-3 | 1.63e-2 ± 2.15e-3 | 1.78e-2 ± 5.90e-3 |
> | Wave2d-MS         | 5.71e-2 ± 1.77e-3 | 6.03e-2 ± 3.04e-3 | 5.58e-2 ± 2.92e-3 | 5.43e-2 ± 5.11e-3 |
>
> Different learning rates (Adam optimizer, $\beta_1=0.9,\beta_2=0.999$; the problem is Poisson2d-MS; 3 random trails):
>
> | Metric (mean ± std) | $\eta=1\times 10^{-4}$ | $\eta=3\times 10^{-4}$ | $\eta=1\times 10^{-3}$ | $\eta=3\times 10^{-3}$ |
> | ------------------- | ---------------------- | ---------------------- | ---------------------- | ---------------------- |
> | MAE                 | 8.37e-2 ± 5.89e-4      | 8.40e-2 ± 8.52e-4      | 8.57e-2 ± 3.28e-3      | 8.56e-2 ± 4.66e-3      |
> | MSE                 | 2.71e-2 ± 2.36e-4      | 2.72e-2 ± 2.05e-4      | 2.75e-2 ± 1.36e-3      | 2.75e-2 ± 1.11e-3      |
> | L1RE                | 4.72e-2 ± 3.40e-4      | 4.74e-2 ± 4.97e-4      | 4.83e-2 ± 1.89e-3      | 4.83e-2 ± 2.65e-3      |
> | L2RE                | 6.34e-2 ± 2.83e-4      | 6.36e-2 ± 2.49e-4      | 6.39e-2 ± 1.53e-3      | 6.39e-2 ± 1.28e-3      |
>
> Different betas $\beta_1, \beta_2$ (Adam optimizer, $\eta=1\times 10^{-3}$; the problem is Poisson2d-MS; 3 random trails):
>
> | Metric (mean ± std) | $(0.9,0.9)$       | $(0.9,0.99)$      | $(0.9,0.999)$     | $(0.99,0.99)$     | $(0.99,0.999)$    |
> | ------------------- | ----------------- | ----------------- | ----------------- | ----------------- | ----------------- |
> | MAE                 | 8.45e-2 ± 8.18e-4 | 8.49e-2 ± 1.25e-3 | 8.57e-2 ± 3.28e-3 | 8.34e-2 ± 2.87e-4 | 8.39e-2 ± 3.86e-4 |
> | MSE                 | 2.74e-2 ± 4.64e-4 | 2.76e-2 ± 5.25e-4 | 2.75e-2 ± 1.36e-3 | 2.75e-2 ± 8.16e-5 | 2.77e-2 ± 9.43e-5 |
> | L1RE                | 4.76e-2 ± 4.50e-4 | 4.79e-2 ± 7.26e-4 | 4.83e-2 ± 1.89e-3 | 4.71e-2 ± 1.63e-4 | 4.73e-2 ± 2.16e-4 |
> | L2RE                | 6.37e-2 ± 5.56e-4 | 6.39e-2 ± 6.18e-4 | 6.39e-2 ± 1.53e-3 | 6.39e-2 ± 1.25e-4 | 6.41e-2 ± 9.43e-5 |
>
> Different number of hidden neural neurons in each layer (the number of hidden layers is 5; the problem is Poisson2d-MS; 3 random trails):
>
> | Metric (mean ± std) | $32$              | $64$              | $128$             | $256$             | $512$             |
> | ------------------- | ----------------- | ----------------- | ----------------- | ----------------- | ----------------- |
> | MAE                 | 8.42e-2 ± 3.77e-4 | 8.38e-2 ± 2.36e-4 | 8.60e-2 ± 3.07e-3 | 8.84e-2 ± 2.05e-3 | 8.49e-2 ± 8.01e-4 |
> | MSE                 | 2.72e-2 ± 1.89e-4 | 2.73e-2 ± 2.94e-4 | 2.80e-2 ± 1.01e-3 | 2.90e-2 ± 8.38e-4 | 2.75e-2 ± 1.89e-4 |
> | L1RE                | 4.75e-2 ± 2.16e-4 | 4.73e-2 ± 1.41e-4 | 4.85e-2 ± 1.75e-3 | 4.99e-2 ± 1.13e-3 | 4.79e-2 ± 4.50e-4 |
> | L2RE                | 6.36e-2 ± 2.36e-4 | 6.36e-2 ± 3.30e-4 | 6.44e-2 ± 1.16e-3 | 6.56e-2 ± 9.63e-4 | 6.38e-2 ± 2.36e-4 |
>
> Different number of hidden layers (the number of hidden neural neurons in each layer is 128; the problem is Poisson2d-MS; 3 random trails):
>
> | Metric (mean ± std) | $3$               | $4$               | $5$               | $6$               | $7$               |
> | ------------------- | ----------------- | ----------------- | ----------------- | ----------------- | ----------------- |
> | MAE                 | 8.39e-2 ± 6.55e-4 | 8.37e-2 ± 8.29e-4 | 8.84e-2 ± 2.05e-3 | 8.21e-2 ± 4.64e-4 | 8.43e-2 ± 4.50e-4 |
> | MSE                 | 2.72e-2 ± 1.41e-4 | 2.70e-2 ± 2.87e-4 | 2.90e-2 ± 8.38e-4 | 2.56e-2 ± 2.36e-4 | 2.73e-2 ± 4.71e-5 |
> | L1RE                | 4.74e-2 ± 3.68e-4 | 4.72e-2 ± 4.64e-4 | 4.99e-2 ± 1.13e-3 | 4.63e-2 ± 2.49e-4 | 4.75e-2 ± 2.49e-4 |
> | L2RE                | 6.35e-2 ± 1.41e-4 | 6.33e-2 ± 2.87e-4 | 6.56e-2 ± 9.63e-4 | 6.17e-2 ± 3.30e-4 | 6.36e-2 ± 9.43e-5 |
>
> We hope this additional information addresses your concern and demonstrates the thoroughness of our approach.

---

> ### Author Response · Authors · 2023-11-18
> **Part III (Q4-6)**
>
> ### Q4: How does the condition number help diagnose and rectify training pathologies in PINNs? Any theoretical analysis or empirical evidence?
>
> In **Corollary 3.4** (in the latest manuscript), we demonstrate a theoretical link between the condition number and error control in PINNs, indicating that a lower condition number usually results in higher accuracy. This correlation is empirically supported by our experiments shown in **Figure 2(b)**, where a strong relationship (with $R^2 > 0.9$) between the condition number and the error of PINNs is evident across three practical problems. Furthermore, **Theorem 3.5** and **Theorem 3.6** establishe that the condition number significantly impacts the convergence of PINNs. This is empirically validated in **Figure 2(c)**, which illustrates that PINNs with lower condition numbers converge more rapidly.
>
> Additionally, our preconditioned PINN, as evidenced in **Table 2**, not only achieves state-of-the-art results in most tested problems but also demonstrates superior performance, reducing errors by an order of magnitude or being the only method to significantly reduce errors below 100% in **7** problems.
>
> Our findings collectively underscore the critical role of the condition number in both diagnosing and improving the training efficacy and accuracy of PINNs.
>
> ### Q5: What is the significance of the proposed approach for solving complex partial differential equations (PDEs)?
>
> Our proposed approach significantly enhances the performance of Physics-Informed Neural Networks (PINNs) in solving complex PDEs. The condition number, particularly in complex PDE scenarios, is often very high, which directly impacts the accuracy and convergence efficiency of PINNs. As demonstrated in **Table 2** of our paper, conventional PINN methodologies exhibit limitations in complex PDE cases like Poisson2d-MS, Heat2d-LT, and Wave2d-MS.
>
> In stark contrast, our preconditioning-based method not only improves error margins but does so dramatically, often by an order of magnitude. This is a crucial advancement, as we've managed to bring error rates consistently below 100% in scenarios where previous baselines failed. Such results underscore the vital role of our approach in resolving intricate PDE challenges, pointing to a substantial leap forward in the applicability and robustness of PINNs in complex mathematical modeling.
>
> ### Q6: What are the implications of the proposed approach for computational efficiency and scalability?
>
> Our research demonstrates notable improvements in computational efficiency with the proposed method, as evidenced by **Figure 3(a)**. This enhancement stems from utilizing a discretized physics loss, which circumvents the extensive computational demands typically associated with Autograd in continuous physics loss calculations. Notably, the additional computational overhead introduced by our preconditioning algorithm is minimal, typically under **3** seconds, and therefore does not significantly impact the overall training time.
>
> Regarding scalability, **Figure 3(b)** illustrates that our method exhibits a more favorable scaling of computation time relative to problem size. This finding suggests that our approach holds substantial promise for application in large-scale problems, potentially overcoming some limitations of traditional methods in this domain.
>
> Our results, therefore, indicate that the proposed approach could be a significant step forward in both the efficiency and applicability of PINNs in solving complex partial differential equations.

---

> ### Author Response · Authors · 2023-11-21
> **Sincerely Seeking Your Valuable Feedback**
>
> Dear Reviewer,
>
> As we approach the final stages of the review process, we would like to express our gratitude for your insightful comments and suggestions. In response, we have thoroughly revised our manuscript, incorporating several key changes and extending the scope of our research.
>
> In line with your recommendations, we have undertaken comprehensive ablation studies. These include detailed comparisons with alternative preconditioning algorithms and in-depth analyses of our method's sensitivity to various hyperparameters and initialization techniques. Our goal with these additional studies is to bolster the validity and generalizability of our findings.
>
> Your expertise has been instrumental in guiding these enhancements, and we are grateful for your valuable input. We are eager to hear your thoughts on the revisions and are open to further discussions. Should you have additional suggestions, we are fully prepared to consider and implement them.
>
> We look forward to your feedback and hope that our revised manuscript meets the high standards expected by the conference.
>
> Warm regards,
>
> Authors

---

### Official Review · Reviewer_8B8P · 2023-10-31

**Soundness:** 1 poor
**Presentation:** 2 fair
**Contribution:** 2 fair
**Rating:** 3
**Confidence:** 3

**Summary:**

This paper proposes a way to alleviate the ill-conditioning of pinns training, proposing a metric to assess ill-conditioning. They provide theoretical results on the relationship between their metric and converge of the method, as well as numerical results illustrating how their approach behaves  on a PINN benchmark.

**Strengths:**

The paper is clear and reads well.

**Weaknesses:**

I have lots of concerns regarding the correctness and depth of the mathematical results.

First off, the 'condition number' you define looks nothing like a condition number. This is a well defined concept in the literature, it is not possible to define it however you like. Moreover, the supremum is taken on a dependant variable so it is not clear for me what is actually varying here.

The central theorem 3.6, which connects their 'condition number' to how the neural network approaches the solution looks nothing like a result one would obtain in the linear case. If one takes a look at the proof in the appendix, we can see how the condition number of the problem is artificially put in the theorem, making the theorem entirely vacuous. Not to mention the strenuous assumptions.

In section 4, "Training PINNs with a Preconditioner" they do not use a NN, $u$ just becomes a vector that is learned: no autograd, just finite differences, no neural networks. This formulation of the problem is very similar to finite elements, and using ILU preconditioning in this context is not new.

Eq 11. Have you simply replaced u_theta with the target u to obtain the result you were expecting? I could be wrong but i think this is a mistake.

Many more flaws in the paper, but this is already sufficient for me to advise for a clear rejection.

**Questions:**

I have no questions at the moment.

---

> ### Author Response · Authors · 2023-11-18
> **Part I (Q1-3)**
>
> Dear reviewer 8B8P,
>
> Thank you for your insightful questions regarding several important problems. Here are our responses to address your concerns, which consist of two parts: Part I (Q1-3) and Part II (Q4-5).
>
> ### Q1: Definition of Condition Number
>
> "The 'condition number' you define looks nothing like a condition number" is a **factual error**. According to [1], the general condition number is defined to be:
>
> > To state the first one we need to define the relative condition number of the mapping $g$ at $x$ as
> > $$
> > \kappa_{\text{rel}}(g,x)\equiv \limsup_{\delta x \rightarrow 0} \frac{\\| g(x+\delta x)-g(x) \\|/\\| g(x) \\|}{\\| \delta x \\| / \\| x \\|}
> > $$
>
> Our definition is not an arbitrary creation but rather a thoughtful extension of the general condition number concept to the specific domain of Physics-Informed Neural Networks (PINNs). Delving into **Definition 3.1**, we can see our definition exactly matches the requirement: compare $\delta x$ with $\delta f$ and $g(x+\delta x)-g(x)$ with $\delta u$. Besides, we clarify that supremum is not taken on any dependent variable but on the trainable parameters $\boldsymbol{\theta}$. We apologize that in **Definition 3.1**, we have neglected $\boldsymbol{\theta}\in \Theta$ for simplicity. To ensure there is no ambiguity, we have added this notation in our revised manuscript.
>
> While the general definition of the condition number provides a broad framework, it is often necessary to tailor this concept to specific domains for practical application. This practice is not uncommon, as seen in matrix theory and root-finding algorithms. Our paper introduces a specialized definition of the condition number for PINNs, aimed at addressing and rectifying their unique training pathologies. This approach is both theoretically and empirically validated in **Sections 5.2** and **5.3** of our paper, demonstrating its relevance and effectiveness in improving the accuracy and convergence of PINNs. We assert that the condition number's ill-conditioned properties in PDEs are a critical factor affecting the performance of PINNs. This is substantiated by our comparative analyses (refer to **Table 2** and **Figure 3(c)**) and highlights the significance of our contribution to the PINN community. We believe that our findings offer a valuable direction for future research in this field.
>
> [1] Demmel, J. W. (1987). On condition numbers and the distance to the nearest ill-posed problem. *Numerische Mathematik*, *51*, 251-289.
>
> ### Q2: The central theorem 3.6, ..., looks nothing like a result one would obtain in the linear case.
>
> "Theorem 3.6 should look like a result in the linear case" is **factually incorrect**. The neural network has non-linear activation functions, whose optimization should not have the same convergence as the linear optimization. Specifically, linear optimization has well-established and predictable convergence conditions while the global convergence of NNs cannot be assured. Furthermore, **Figure 2(c)** demonstrates that as the condition number decreases, there is a corresponding acceleration in convergence, empirically supporting the validity of our theorem.
>
> ### Q3: "Training PINNs with a Preconditioner" they do not use a NN.
>
> "Training PINNs with a Preconditioner they do not use a NN" is a **factual error**. In **Line 7** of **Algorithm 1**, we have explicitly declared the use of the neural network. Furthermore, in **Appendix D.1**, we provide detailed information regarding the architecture and hyperparameters of the neural network used, which further affirms its integral role in our work. Our model is a physics-informed neural network with a preconditioned discretized loss, fundamentally differentiating it from traditional FEM approaches.
>
> To further clarify, $u_{\\boldsymbol{\\theta}}$ is a neural network that takes a coordinate $\\boldsymbol{x}$ as input and outputs a scalar for prediction and the vector $\\boldsymbol{u}_{\\boldsymbol{\\theta}}$ is not a static vector but the neural network outputs at the grid locations:
>
> $$
> (u_{\\boldsymbol{\\theta}}(\\boldsymbol{x}^{(i)}))_{i=1}^N.
> $$
>
> Then, we incorporate the network outputs $\\boldsymbol{u}_{\\boldsymbol{\\theta}}$ into the preconditioned loss function (refer to **Eq. (12)**).
>
> We do not optimize the vector $\\boldsymbol{u}_{\\boldsymbol{\\theta}}$ directly but the trainable parameters $\\boldsymbol{\theta}$ of the neural network.

---

> ### Author Response · Authors · 2023-11-18
> **Part II (Q4-5)**
>
> ### Q4: Question about Eq. (11).
>
> We appreciate the reviewer’s attention to the details in our manuscript and apologize for any confusion caused by the initial presentation of Eq. (11). In response to your concerns, we have provided a detailed and rigorous mathematical derivation in **Appendix B.1** of the revised manuscript.
>
> The statement that "simply replaced u_theta with the target u" is **factually incorrect**. If such a substitution were made in **Eq. (4)** of **Definition 3.1**, it would lead to a meaningless "0/0" situation.
>
> We encourage the reviewer to examine the revised section in **Appendix B.1** for a comprehensive understanding of the derivation and its implications. We are confident that this additional information will clarify any misunderstandings and demonstrate the correctness of our approach.
>
> ### Q5: Many other flaws.
>
> Thank you for your review. We note your comment about 'many more flaws' in our paper. To ensure we can address these effectively, could you please provide specific details about these additional concerns? Your insights are crucial for us to improve and refine our manuscript. We appreciate your time and expertise in guiding our revisions.

---

> > ### Comment · Reviewer_8B8P · 2023-11-20
> >
> > Q1.
> > A condition number measures the sensitivity of a solution wrt to pertubations in the inputs, and is always defined with respect to a problem. For example, the condition number of a matrix is with respect to inversion, a relative change in some matrix $A$ of norm $\epsilon$ (small) may change $A^{-1}$ by $\epsilon\kappa(A)$, $\kappa(A)$ being the condition number.
> >
> > I am assuming that the condition number defined in the paper is with respect to the training problem. Hence, in order to motivate your definition of condition number, we would expect a result that tightly links the condition number to convergence of the network. However, like I have mentioned in my original review, the result of 3.6 is vacuous as you could have added any scalar instead of your condition number as you take it back out again in the proof. So then what tells me that this quantity is the correct one? In the case where the solution u is linear (eg galerkin discretization) what does the condition number reduce to?
> >
> > Supremum over dependant variable.  My concern was that by taking the supremum over the dependant variable you had to make an additional assumption of expressivity of the NN. I see this assumption has now been added in the main text. In general it would be nice if the assumptions would be mentioned in the premise of the theoretical results, not in the proofs.
> >
> > Q3. No, what I said is not factually incorrect. I am talking about your condition number, which does not take into account the fact that you are using an NN (nor PINNs, as you are doing finite differences). If you want to adhere to the notion of condition number discussed in [1], since you are taking the supremum wrt to the parameters, shouldn’t this imply that the inputs of your problem are the parameters (and not the residual)? I am asking this because as a consequence of this, your condition number is totally oblivious to the architecture of the NN, which seems quite peculiar.
> >
> > Q4. Thank you for the update.

---

> ### Author Response · Authors · 2023-11-21
> **Thank you for your feedback (Part I)**
>
> ## Q1: Supremum over dependent variable.
>
> We believe your concern has been addressed since the assumption is explicitly mentioned in **Definition 3.1**.
>
> ## Q2: Definition of condition number.
>
> It seems that we have reached a consensus: **the condition number we defined meets the definition of condition number**. The main disagreement now is whether the problem output should be the predicted solution or the optimal network parameters. We think it should be the former (our current choice), based on the following reasons:
>
> 1. The point is that the optimal parameter $\boldsymbol{\theta}^*$ is likely to be **non-unique** as the universal approximation theorem [1] does not ensure the uniqueness. We cannot make the definition well-defined without the uniqueness of $\boldsymbol{\theta}^*$. Specifically, there could exist another $\boldsymbol{\theta}'$ such that $\\| \boldsymbol{\theta}' - \boldsymbol{\theta}^* \\|$ is non-zero but $u_{\boldsymbol{\theta}'}$ already approaches $u$.
>
> 2. What we really care about in the PINN problem is **how well the neural network $u_{\boldsymbol{\theta}}$ approximates the solution $u$** instead of how close between $\boldsymbol{\theta}$ and $\boldsymbol{\theta}^*$ is. As discussed in [2], in different random trials, we are likely to obtain different parameters that are far from each other but all well approximate $u$. It is clearly unreasonable to measure $\\| \boldsymbol{\theta} - \boldsymbol{\theta}^* \\|$ in this case.
>
> 3. Empirically, in **Figure 2(b)** and **2(c)**, we can find that the current condition number can significantly affect the accuracy and convergence of PINNs. Furthermore, **our ablation study** shows that the NN architecture plays a minimal role in the performance of PINNs.
>
>    Different number of hidden neural neurons in each layer (the number of hidden layers is 5; the problem is Poisson2d-MS; 3 random trails):
>
>    | Metric (mean ± std) | $32$              | $64$              | $128$             | $256$             | $512$             |
>    | ------------------- | ----------------- | ----------------- | ----------------- | ----------------- | ----------------- |
>    | MAE                 | 8.42e-2 ± 3.77e-4 | 8.38e-2 ± 2.36e-4 | 8.60e-2 ± 3.07e-3 | 8.84e-2 ± 2.05e-3 | 8.49e-2 ± 8.01e-4 |
>    | MSE                 | 2.72e-2 ± 1.89e-4 | 2.73e-2 ± 2.94e-4 | 2.80e-2 ± 1.01e-3 | 2.90e-2 ± 8.38e-4 | 2.75e-2 ± 1.89e-4 |
>    | L1RE                | 4.75e-2 ± 2.16e-4 | 4.73e-2 ± 1.41e-4 | 4.85e-2 ± 1.75e-3 | 4.99e-2 ± 1.13e-3 | 4.79e-2 ± 4.50e-4 |
>    | L2RE                | 6.36e-2 ± 2.36e-4 | 6.36e-2 ± 3.30e-4 | 6.44e-2 ± 1.16e-3 | 6.56e-2 ± 9.63e-4 | 6.38e-2 ± 2.36e-4 |
>
>    Different number of hidden layers (the number of hidden neural neurons in each layer is 128; the problem is Poisson2d-MS; 3 random trails):
>
>    | Metric (mean ± std) | $3$               | $4$               | $5$               | $6$               | $7$               |
>    | ------------------- | ----------------- | ----------------- | ----------------- | ----------------- | ----------------- |
>    | MAE                 | 8.39e-2 ± 6.55e-4 | 8.37e-2 ± 8.29e-4 | 8.84e-2 ± 2.05e-3 | 8.21e-2 ± 4.64e-4 | 8.43e-2 ± 4.50e-4 |
>    | MSE                 | 2.72e-2 ± 1.41e-4 | 2.70e-2 ± 2.87e-4 | 2.90e-2 ± 8.38e-4 | 2.56e-2 ± 2.36e-4 | 2.73e-2 ± 4.71e-5 |
>    | L1RE                | 4.74e-2 ± 3.68e-4 | 4.72e-2 ± 4.64e-4 | 4.99e-2 ± 1.13e-3 | 4.63e-2 ± 2.49e-4 | 4.75e-2 ± 2.49e-4 |
>    | L2RE                | 6.35e-2 ± 1.41e-4 | 6.33e-2 ± 2.87e-4 | 6.56e-2 ± 9.63e-4 | 6.17e-2 ± 3.30e-4 | 6.36e-2 ± 9.43e-5 |
>
>    Therefore, we think it is reasonable not to incorporate the NN architecture in the definition of condition number but to focus on the underlying physical challenges.
>
> [1] Funahashi, K. I. (1989). On the approximate realization of continuous mappings by neural networks. *Neural networks*, *2*(3), 183-192.
>
> [2] Sagun, L., Evci, U., Guney, V. U., Dauphin, Y., & Bottou, L. (2017). Empirical analysis of the hessian of over-parametrized neural networks. *arXiv preprint arXiv:1706.04454*.
>
> ## Q3: Can the condition number be replaced by any scalar in Theorem 3.6?
>
> No, this statement is **incorrect**. Without further a priori, the condition number is **the smallest constant** that makes **Eq. (10)** (see **Theorem 3.5** in the latest manuscript) hold. In the proof (see **Appendix A.5**), we plug **Eq. (51)** into **Eq. (53)** to obtain the final result, where the condition number is the smallest constant that makes **Eq. (53)** hold due to the definition of the supremum.
>
> ## Q4: In the case where the solution u is linear, what does the condition number reduce to?
>
> According to **Definition 3.1**, the definition of the condition number is irrelevant to whether the solution $u=u(\boldsymbol{x})$ is linear with respect to the input coordinate $\boldsymbol{x}$.

---

> ### Author Response · Authors · 2023-11-22
> **Thank you for your feedback (Part II)**
>
> ## Introducing an additional  theorem
>
> In response to your valuable feedback, we have expanded our theoretical analysis. We introduce a new theorem in the revised manuscript (now in **Theorem 3.6**) that investigates the global convergence of PINNs under the infinite-width assumption. This analysis utilizes the neural tangent kernel (NTK) framework to establish a link between the condition number and the rate of global convergence, where the factor of the neural network is also considered. We believe this addition addresses your concerns and enriches the paper significantly. The detailed proof is presented in **Appendix A.7**. We invite you to review this new theorem and look forward to your thoughts.

---

> > ### Comment · Reviewer_8B8P · 2023-11-23
> >
> > Thank you for your update and the additional experiments.
> >
> > Just to be clear--I do not think we have reached a consensus as I did not change my initial thinking, your condition number looks like some quantity that measures the stability of the PDE, not the learning problem, e.g. see https://arxiv.org/pdf/2006.16144.pdf (Section 2.4 An abstract estimate on the generalization error), where C_pde would corresponds to your condition number (which actually requires having access to the solution in order to compute the condition number?).
> >
> > Moreover, i do not think uniqueness is indispensable to define a condition number, e.g. the condition number could be defined locally: this is what is done in the non-linear setting. Furthermore, in the context of time-dependent problems, the paper diverges from their initial preconditioning approach, opting additionally for the method described in Wang et al. While I only recently noticed this shift, given the tight deadline, I will not consider it in my evaluation.
> >
> > Lastly, the additional convergence result is done in continuous time, the condition number does not make sense in this setting as time can always be reparametrized by increasing the learning rate, thus accelerating training.
> >
> > As is, I would not recommend for acceptance. The mathematical results are not very convincing, more importantly it feels like the correct mathematical framework and has not been well captured (for reasons above). However, i could be mistaken.

---

> > > ### Author Response · Authors · 2023-11-23
> > > **Thank you for your hard work in reviewing**
> > >
> > > ## Q1: I do not think uniqueness is indispensable to defining a condition number.
> > >
> > > Actually, you will yield an **ill-defined condition number** if the uniqueness is missing, which is why we do not recommend defining the problem output to be optimal parameters $\boldsymbol{\theta}^*$. Specifically, here are our reasons:
> > >
> > > 1. Suppose we have multiple $\\boldsymbol{\\theta}^*$. As discussed in [1], in different random trails, you are likely to converge to different $\\boldsymbol{\\theta}^*$, which means that the value of **condition number will depend on the random seed** you chosen (and the random initialization scheme). Obviously, this is unreasonable.
> > > 2. Even if you define the condition number locally, due to the randomness of the neural network's training, you do not know which local one corresponds to a specific random trail. And thus, the condition number becomes elusive. Although we can define a "distribution" of condition numbers, this is beyond our scope and could be a further direction.
> > > 3. In **Theorem 3.6**, **Table 2**, and **Figure 2(b)(c)**, our current condition number provides helpful insights into the training pathology of PINNs, validated by our experiments and our method's superior performance over other PINN baselines. Therefore, we consider our current definition to have unique advantages and should be acceptable.
> > >
> > > ## Q2: In the context of time-dependent problems, the paper diverges from their initial preconditioning approach.
> > >
> > > We refer the reviewer to **Line 8 in Algorithm 3**. The preconditioner is **explicitly used** to improve the training of PINNs. However, we agree with the reviewer that maybe we should perform more ablation studies on time-dependent problems. Due to the tight deadline, we leave this work to the future.
> > >
> > > ## Q3: The additional convergence result is done in continuous time, the condition number does not make sense in this setting as time can always be reparametrized by increasing the learning rate, thus accelerating training.
> > >
> > > Thank you for your additional time in reviewing our newly added theorem. However, we should explicitly point out that the reviewer's claim is **factually incorrect**, due to the following reasons:
> > >
> > > 1. According to **Definition 3.1**, the definition of the condition number does not depend on the time of gradient flow. Therefore, the gradient flow will not make the condition number meaningless.
> > > 2. According to [2], the learning rate in NTK is **infinitely small**. So increasing the learning rate to accelerate training is impossible. Besides, in NTK (see [3]), the convergence rate depends on the eigenvalues of the kernel, which converge to constant when the width approaches infinity and cannot be increased by parametrizing the time. Moreover, in **Theorem 3.6**, we show that the eigenvalues are related to the condition number, therefore supporting our claim that the condition number can affect the convergence of PINNs.
> > >
> > > ## Q4: More questions on the condition number.
> > >
> > > 1. Does C_pde correspond to your condition number?
> > >
> > >    No, according to **Assumption 2.1** in your reference paper, $C_{pde}$ explicitly depends on the test functions (or input functions), while our condition number is an inherent property of the problem and does not have this limitation.
> > >
> > > 2. Do we actually require having access to the solution in order to compute the condition number?
> > >
> > >    According to **Section 5.2** in our paper, we can approximate the condition number even without access to the real solution.
> > >
> > > [1] Sagun, L., Evci, U., Guney, V. U., Dauphin, Y., & Bottou, L. (2017). Empirical analysis of the hessian of over-parametrized neural networks. *arXiv preprint arXiv:1706.04454*.
> > >
> > > [2] Jacot, A., Gabriel, F., & Hongler, C. (2018). Neural tangent kernel: Convergence and generalization in neural networks. *Advances in neural information processing systems*, *31*.
> > >
> > > [3] Wang, S., Yu, X., & Perdikaris, P. (2022). When and why PINNs fail to train: A neural tangent kernel perspective. *Journal of Computational Physics*, *449*, 110768.

---

### Official Review · Reviewer_MgQM · 2023-11-01

**Soundness:** 4 excellent
**Presentation:** 3 good
**Contribution:** 3 good
**Rating:** 6
**Confidence:** 3

**Summary:**

This paper defines the condition number as a metric for the difficulty and ill-posedness of the PINN training problem. The main argument is that the condition number is problem-dependent and almost algorithm-independent, therefore the problem must be alleviated by pre-codnitioning. They also support their theory with numerical examples on 16 PDE problems.

**Strengths:**

1. The idea and the theoretical standing of the paper is sound; the pre-conditioning of linear solvers has been studied for decades in numerical computation literature and is proven effective.

2. The suite of experiments does support the hypotheses made in the introduction under many difficulty conditions such as time-dependency, non-linearity, irregular geometry, discontinuity, etc.

3. The research and writing sequence is right; the work does start by defining and making the case for an underlying problem, showing that it exists in the first place, and then proposes a solution.

3. The implementation is made available and the work includes a detailed appendix.

**Weaknesses:**

1. The main turn-off of the work is its lack of scalability to higher-dimensional PDEs. One major part of the "complex problems" defined in Hao et al. (2022) is its high-dimensional PDE category, which seems left behind altogether in this work. The key advantage and promise of PINNs, compared to mesh-based/FEM/etc counter-parts, is their ability to generalize to higher-dimensional problems. This ability is unfortunately tainted by the need for the creation of a mesh in the proposed solution. On the second page, the authors make the assumption of d<=4 for the practicality of the proposed solution. This is a significant limitation of the work, and narrows its applicability to situations where FEMs can be expected to already be applicable.

2. Most of the experiments in the paper are done with 5 random seeds. This sample size is inadequate for reliable and significant conclusions.

3. The Poisson example in Figure 2a is overly simplistic. Since this is the basis for the paper, I wish the underlying problem was a bit more challenging and representative of the real-world problems than a 1-d Poisson problem.

**Questions:**

1. In Figure 1.a, why does the iterations axes start at 500? It looks like PCPINN is getting a head-start, which seems unfair to the other methods.

2. Figure 2b only shows 3 PDE problems of the Wave, Burgers, and the Helmholtz equations. Could the authors explain why more of the 16 considered problems were not included in this experiment?

3. Here are a few typos in the paper:

    * In the title of Section 3.2, "Condition" is misspelled as "Contion".

    * On Page 5, there is a minor grammatical error two lines under Equation 9: "closely enough" should better be replaced with "close enough".

---

> ### Author Response · Authors · 2023-11-18
> **Part I (Q1-3)**
>
> Dear reviewer MgQM,
>
> Thank you for your valuable feedback on improving our paper. Here are our responses to address your concerns, which consist of two parts: Part I (Q1-3) and Part II (Q4-6).
>
> ### Q1: Scalability to higher-dimensional PDEs.
>
> We appreciate the concern regarding the scalability of our method to higher-dimensional PDEs. While we recognize the limitation of our current approach to dimensions $d\le 4$, as discussed in **Section 5.3**, this focus aligns with the practicality of addressing many real-world phenomena, which predominantly exist in three-dimensional space.
>
> Furthermore, as illustrated in **Figure 3(b)**, our method demonstrates promising scalability with respect to mesh size. This suggests potential superiority over traditional methods like FEM for larger problems within the $d\le 4$ space.
>
> Significantly, our method excels in addressing **inverse problems** [1,2], a critical area where physics-informed neural networks have shown distinct advantages over traditional methods. Our studies on two inverse problems from the benchmark [3], as outlined in the table provided, exhibit not only improvements over state-of-the-art PINN baselines but also notable enhancements in accuracy compared to the adjoint method. This is particularly relevant as numerical methods often struggle with noise sensitivity in inverse problems. And the improved performance underscores the method's potential despite its current dimensional limitations.
>
> | Problem         | L2RE (Ours)           | L2RE (SOTA)       | L2RE (Adjoint)    | Time (Ours) | Time (SOTA) | Time (Adjoint) |
> | --------------- | --------------------- | ----------------- | ----------------- | ----------- | ----------- | -------------- |
> | Poisson Inverse | **1.80e-2 ± 9.30e-3** | 2.45e-2 ± 1.03e-2 | 7.82e+2 ± 0.00e+0 | 1.87e+2     | 4.90e+2     | 1.40e+0        |
> | Heat Inverse    | **9.04e-3 ± 2.34e-3** | 5.09e-2 ± 4.34e-3 | 1.50e+3 ± 0.00e+0 | 3.21e+2     | 3.39e+3     | 1.07e+1        |
>
> Our ongoing research aims to extend the applicability of our method to higher-dimensional PDEs with neural-based preconditioners, and we value the feedback to guide our future work.
>
> [1] Karniadakis, G. E., Kevrekidis, I. G., Lu, L., Perdikaris, P., Wang, S., & Yang, L. (2021). Physics-informed machine learning. *Nature Reviews Physics*, *3*(6), 422-440.
>
> [2] Raissi, M., Perdikaris, P., & Karniadakis, G. E. (2019). Physics-informed neural networks: A deep learning framework for solving forward and inverse problems involving nonlinear partial differential equations. *Journal of Computational physics*, *378*, 686-707.
>
> [3] Hao, Z., Yao, J., Su, C., Su, H., Wang, Z., Lu, F., ... & Zhu, J. (2023). PINNacle: A Comprehensive Benchmark of Physics-Informed Neural Networks for Solving PDEs. *arXiv preprint arXiv:2306.08827*.
>
> ### Q2: The Poisson example in Figure 2(a) is overly simplistic.
>
> Thank you for highlighting the Poisson example in our paper. It's important to clarify that this example is primarily for illustration and not the central focus of our work. We chose a 1D Poisson problem for its analytical simplicity, which aids in clearly demonstrating the behavior and existence of the condition number. This simplicity allows for an analytical expression of the condition number, facilitating a more straightforward discussion and understanding.
>
> We acknowledge that a more complex problem, like a 2D Poisson equation, might be more representative of real-world challenges. However, such complexities could obscure the fundamental principles we aim to illustrate, especially where an analytical expression of the condition number is not feasible.
>
> Our main contributions, as detailed in the paper, are grounded in the two theorems that link the condition number to the error and convergence in PINNs and the comprehensive experiments across a range of problems. These elements of our work address a spectrum of complexities (refer to **Table 1**) and are not confined to simple PDEs. They illustrate the applicability and robustness of our approach in diverse real-world scenarios, as also acknowledged in the strengths section of your review.
>
> ### Q3: In Figure 1(a), why does the iteration axis start at 500?
>
> Thank you for your insightful observation regarding Figure 1(a). Upon re-examination, we realized an inadvertent misalignment in the training histories of the compared methods, leading to the omission of the initial iterations. We appreciate you bringing this to our attention. We have rectified this in the updated version of the manuscript to ensure accurate and fair comparison, and have meticulously rechecked the data to prevent similar oversights. The revised **Figure 1(a)** now accurately reflects the complete training process from the outset.

---

> ### Author Response · Authors · 2023-11-18
> **Part II (Q4-6)**
>
> ### Q4: Inadequate random samples.
>
> Thank you for pointing out the concern regarding the sample size in our experiments. We have considered your feedback and re-evaluated all experiments using 10 random trials. To succinctly demonstrate the consistency and reliability of our findings, we compared the outcomes of the 5-trial and 10-trial experiments. Our findings show that the results from the 10-trial evaluations align closely with those from the original 5-trial tests, indicating that our initial conclusions are consistent and reliable. Moreover, the comparison with the state-of-the-art (SOTA) baseline methods remains unchanged, affirming the robustness of our approach.
>
> We appreciate your feedback as it has helped us strengthen the validity of our research, and we are confident that the additional results further establish the reliability of our findings.
>
> | L2RE (mean ± std) | 5 Random Samples  | 10 Random Samples | Best Baseline     |
> | ----------------- | ----------------- | ----------------- | ----------------- |
> | Burgers1d-C       | 1.42e-2 ± 1.62e-4 | 1.41e-2 ± 2.16e-4 | 1.43e-2 ± 1.44e-3 |
> | Burgers2d-C       | 5.23e-1 ± 7.52e-2 | 4.90e-1 ± 2.94e-2 | 2.60e-1 ± 5.78e-3 |
> | Poisson2d-C       | 3.98e-3 ± 3.70e-3 | 1.84e-3 ± 9.18e-4 | 1.23e-2 ± 7.37e-3 |
> | Poisson2d-CG      | 5.07e-3 ± 1.93e-3 | 5.04e-3 ± 1.53e-3 | 1.43e-2 ± 4.31e-3 |
> | Poisson3d-CG      | 4.16e-2 ± 7.53e-4 | 4.13e-2 ± 5.08e-4 | 1.02e-1 ± 3.16e-2 |
> | Poisson2d-MS      | 6.40e-2 ± 1.12e-3 | 6.42e-2 ± 7.62e-4 | 5.90e-1 ± 4.06e-2 |
> | Heat2d-VC         | 3.11e-2 ± 6.17e-3 | 2.61e-2 ± 3.74e-3 | 2.12e-1 ± 8.61e-4 |
> | Heat2d-MS         | 2.84e-2 ± 1.30e-2 | 2.07e-2 ± 6.52e-3 | 4.40e-2 ± 4.81e-3 |
> | Heat2d-CG         | 1.50e-2 ± 1.17e-4 | 1.55e-2 ± 5.37e-4 | 2.39e-2 ± 1.39e-3 |
> | Heat2d-LT         | 2.11e-1 ± 1.00e-2 | 1.87e-1 ± 8.41e-3 | 9.99e-1 ± 1.05e-5 |
> | NS2d-C            | 1.28e-2 ± 2.44e-3 | 1.21e-2 ± 2.53e-3 | 3.60e-2 ± 3.87e-3 |
> | NS2d-CG           | 6.62e-2 ± 1.26e-3 | 6.36e-2 ± 2.21e-3 | 8.24e-2 ± 8.21e-3 |
> | NS2d-LT           | 9.09e-1 ± 4.00e-4 | 9.09e-1 ± 9.00e-4 | 9.95e-1 ± 7.19e-4 |
> | Wave1d-C          | 1.28e-2 ± 1.20e-4 | 1.28e-2 ± 1.55e-4 | 9.79e-2 ± 7.72e-3 |
> | Wave2d-CG         | 5.85e-1 ± 9.05e-3 | 5.48e-1 ± 8.69e-3 | 7.94e-1 ± 9.33e-3 |
> | Wave2d-MS         | 5.71e-2 ± 5.68e-3 | 6.07e-2 ± 8.20e-3 | 9.82e-1 ± 1.23e-3 |
> | GS                | 1.44e-2 ± 2.53e-3 | 1.44e-2 ± 3.10e-3 | 7.99e-2 ± 1.69e-2 |
> | KS                | 9.52e-1 ± 2.94e-3 | 9.52e-1 ± 3.03e-3 | 9.57e-1 ± 2.85e-3 |
>
> ### Q5:  why more of the 16 considered problems were not included in Figure 2(b)?
>
> Thank you for this insightful question. We carefully considered the scope and feasibility of our experiments when deciding which problems to include in Figure 2(b). Due to the extensive computational resources required (**19-40** instances per problem, each run in **7** random trials), it was not practical to include all 16 problems in this specific experiment.
>
> To ensure a thorough yet efficient analysis, we selected three representative problems that encompass a broad range of characteristics, such as non-linearity and time dependency, where the running time was also considered. These problems were chosen to effectively demonstrate the relationship between the condition number and the accuracy of PINNs, while also maintaining a balance between comprehensiveness and manageability. We believe this approach offers meaningful insight into our method’s capabilities without overextending resources or diluting the focus of our study.
>
> ### Q6: A few typos in the paper.
>
> Thank you for pointing out these typographical errors. We appreciate your attention to detail, as it helps enhance the quality of our manuscript. We have carefully revised and corrected these errors in the updated version of our paper.

---

> ### Author Response · Authors · 2023-11-21
> **Sincerely Seeking Your Valuable Feedback**
>
> Dear Reviewer,
>
> As the review process nears its conclusion, we are reaching out to solicit your thoughts on the revisions made to our manuscript, inspired by your valuable insights. In our pursuit of excellence and scientific rigor, we have not only addressed your initial comments but also extended our research scope.
>
> Specifically, we've conducted in-depth experiments on inverse problems and expanded our analysis over a broader range of random trials. These enhancements aim to more convincingly demonstrate the effectiveness and reliability of our approach. Additionally, we have meticulously corrected the identified typographical errors and rectified the graphical issue in Figure 1(a), as reflected in the updated version of our paper.
>
> Your expertise in this domain has been a guiding light in these improvements, and we deeply appreciate your constructive criticism. We are keen to understand your perspective on these latest modifications. Please rest assured that we remain fully committed to further discussions and are prepared to undertake any additional revisions that you may suggest.
>
> We eagerly anticipate your feedback and hope our efforts align with the high standards of the conference.
>
> Warm regards,
>
> Authors

---

### Meta-Review · Area_Chair_y7V3 · 2023-12-10

**Metareview:**

The paper introduces condition number for PINNS, and introduces preconditioning.
By preconditioning, the authors mean to minimize not the norm of the residual, but the norm of the preconditioned residual (i.e., the regression problem)

Strengths:

1) The problem if conditioning is important

Weaknesses:
1) I might be wrong, but I am almost sure that the paper has been written with a several help of ChatGPT. The sentences like 'We delve into the foundational problem of both mathematics and physics, the Poisson equation'. This all creates unnecessary pathos and bad feeling while reading the paper.
2) The whole idea is computationally not useful: if we have the preconditioner, there is no need for PINNS, and we are just minimizing the L2-norm of the residual, which converges (slightly) better, according to the numerical experiments.
3) The structure of the results and proof is misleading. It mixes condition number and the norm of the inverse operator, for example.

**Justification For Why Not Higher Score:**

No way this paper can be accepted: bad writing, no useful idea, etc.

**Justification For Why Not Lower Score:**

N/A

---

### Decision · Program_Chairs · 2024-01-16

Reject